# Impact of pitch angle fluctuations on airborne lidar sensing ahead along the flight direction

Alexander Sergeevich Gurvich [1] and Victor Alexeevich Kulikov [1,2]

[1]Obukhov Institute of Atmospheric Physics, Russian Academy of Science, 3 Pyzhevskii pereulok str., Moscow, Russia, 119017

[2]University of Dayton, 300 College Park, Dayton, OH, USA 45469

*Correspondence to:* Victor A. Kulikov (victoralexkulikov@gmail.com)

**Abstract.** Airborne lidar sensing ahead along the flight direction can serve for notification of clear air turbulence (CAT) and help to prevent injuries or fatal air accidents. The validation of this concept was presented in the framework of the DELICAT (DEmonstration of LIdar based CAT detection) project. However, the strong variations of signal level sometimes, which were observed during the DELICAT measurements but not explained, indicated the need of a better understanding the observational errors due to geometrical factors. In this paper, we discuss possible error sources imminent to this technique, related to fluctuations of the flight parameters, which may lead to strong signal variations caused by the random deviations of the sensing beam from the forward flight trajectory. We analyze the variations of backscattered lidar signal caused by fluctuations of the most important ahead sensing flight parameter, the pitch angle. The fluctuation values considered in the paper correspond to the error limits of the compensational gyro-platform used in the civil aviation. The part of pitch angle fluctuations uncompensated by the beam steering device in the presence of aerosol concentration variations can lead to noticeable signal variations that can be mistakenly attributed to wind shear, turbulence or fast evolution of aerosol layer. We formulate the criteria that allow the recognition of signal variations caused by pitch angle fluctuations. Influence of these fluctuations is shown to be stronger for aerosol variations on smaller vertical scales. An example of DELICAT observations indicating a noticeable pitch angle fluctuation impact is presented.

## 1 Introduction

Airborne lidar systems (Fukuchi and Shiina, 2012; Weitkamp, 2006) may play a significant role in alarming, preventing, and compensating problems caused by atmospheric turbulence. Such system were previously developed for short range sounding (Schmitt et al., 2007; Jentink and Bogue, 2005). Recently, a medium range lidars were developed, built and tested (Huffaker and Hardesty, 1996; Inokuchi et al., 2009; Veerman et al., 2014; Vrancken et al., 2016; Inokuchi et al., 2009b; Targ et al., 1996; Thales Avionics and ONERA, 2004). One of these systems was developed in the framework of the DELICAT project (DEmonstration of LIdar based Clear Air Turbulence detection) (Veerman et al., 2014; Vrancken et al., 2016). Medium range systems are designed to work up to 20-30 km sensing distance, which corresponds to 2–10 minutes of warning time for typical

flight speed of airplane and helicopter correspondingly. An earlier warning is preferable and airborne lidar with larger sensing distance could be developed in future.

Sensing of turbulence can be based on backscattered signal from air density fluctuations (Veerman et al., 2014; Feneyrou, 2009; Vrancken et al., 2016) which allows detecting turbulence even in the absence of aerosol scatterers. At the same time, dust and smog, water vapor etc. contribute to the backscattered signal as well. The signal filtration is a good method to exclude undesirable contributions. For example, Hair and co-authors used an extremely narrowband iodine vapor (I2) absorption filter to eliminate the aerosol returns and pass the wings of the molecular spectrum (Hair et al., 2008). At the same time, in the DELICAT system the depolarization was used (Vrancken et al. 2016). Backscattered signal measurements at different polarizations (Burton et al., 2015; Veerman et al., 2014) will only allow excluding the component produced by non-spherical aerosol particles. The measured signal is, however, composed of the responses of different atmospheric components which can include the spherical aerosol. The presence of atmospheric aerosol should not be a critical problem for turbulence detection. However, changes of a aerosol layer density during the observation time and the experimental noise which can affect signal in both polarizations simultaneously could be a problem for backscattered signal analysis.

There is another technique of CAT detection based on backscattering enhancement (BSE) effect which was initially found in the theoretical research (Vinogradov et al., 1973) and then experimentally confirmed (Gurvich and Kashkarov, 1977). In framework of DELICAT project the idea of possible turbulence strength estimation based on BSE was theoretically analyzed and reported (Gurvich, 2012; Gurvich and Kulikov, 2013). The two channel scheme based on backscattering enhancement (BSE) looks very promising for future airborne applications in light of both thorough theoretical analysis and experimental evidence of success reported in (Banakh and Smalikho, 2011; Banakh, 2011; Banakh and Razenkov, 2016a, b). This techniqie is also sensitive to the airborne specific noise caused by fluctuations of flight parameters.

The atmospheric effects can bend the sensing beam and prevent to lidar turbulence detection based on any principle. The turbulence anisotropy can noticeable bend the light propagated over such long distances (Gurvich and Chunchuzov, 2003; Sofieva et al., 2010). This impact should be almost negligible for short fifteen km optical path; possible laser beam trajectory deviation of about ten meters is small taking into account the thickness of cluster discussed in our paper (100 meters). At the same time, refractive layers can also significantly change the trajectory of optical wave propagation (Werf, 2003; Nunalee et al., 2015). The consideration of such effects can be performed in the framework of geometrical (Southwell, 1982; Werf, 2003; Nunalee et al., 2015) or wave optics (Vorontsov and Kulikov, 2015; Kulikov et al., 2017). Both turbulence anisotropy and possible impact of refractive layers should be considered in the case of extended sensing distances.

A series of atmospheric processes influence the aerosol concentration and turbulence strength on temporal and spatial scales of medium range sensing. The aerosol concentration can change due to wind shear and evaporation/condensation processes (Ivlev and Dovgalyuk, 1999). For example, small cloud with horizontal characteristic scales about one kilometer can be displaced completely out of originally occupied volume during 40–200 sec by the wind with a speed within the range of 5–25 m/s (Liu et al., 2002). Clouds could be split up into numerous small clusters at the horizontal scale of one or several kilometers. Such splitting was observed for different types of aerosol (Chazette et al., 2012; Cadet et al., 2005; Reichardt et al., 2002). The concentrations of both submicron aerosol and gas may change by 2–3 times during the equilibration process at characteristic

time scales of about 3 minutes (Meng and Seinfeld, 1996). Gravity waves (Nappo, 2013; Fritts and Alexander, 2003) are one of the reasons of CAT (Plougonven and Zhang, 2016; Lane et al., 2003) and new results suggest that turbulence was most strongly forced at the scale of about 700 m (Koch et al., 2005). The smallest spatial and temporal scales of gravity waves amount to about 1 km and 1-2 minutes, respectively (Lu and Koch, 2008; Koch et al., 2005; Plougonven and Zhang, 2016). Therefore,

lidar sensing ahead along the flight direction does not only allow the operational detection of dangerous atmospheric conditions but can also provide information on macrostructures in the aerosol spatiotemporal distribution. At the same time, the signal variations at this time scale may be caused by the variations of lidar sensing trajectory due to the fluctuations of the flight parameters.

Backscattered signal can also be influenced by changing laser pulse properties or atmospheric propagation effects. Laser

instability leads to time variation of both power and shape of pulses, which results in the change of the backscattered signal. Multipath propagation effect is usually ignored in consideration of backscattered signal, which can significantly degrade the accuracy of the measurement analysis (Godbaz et al., 2012). The detectors can be a source of noise, which depends on the input signal (Acharya et al., 2004). These factors also contribute to the complexity of the signal analysis.

In this paper, we discuss the source of errors, which is specific to the airborne measurements. Variations of aircraft flight

height and direction angle are always present in airborne measurements and they influence the observed backscattered signal. Uncontrolled fluctuations of flight height are usually about several meters and lead to the same height shift along the sensing path. It is highly probable that atmospheric aerosol and turbulence properties do not changes noticeably at the scale of a few meters. Variations of flight direction angle lead to variations of the sensing pulse trajectory. Variations of sensing angles for lidars mounted on gyro-platform should be within the error limits of these compensating systems. The accuracy of pitch angle

measurements and fluctuation compensation is about 0.1–0.2 degree rms (SOMAG AG Jena, 2016; Temp-Avia, 2016). Thus the uncompensated angles lie in the range of 0.3–0.6 degrees, which corresponds to 150–300 m shift at the end of a 30 km path. Roll and yaw angle fluctuations do not influence the backscattered signal because this shift is small as compared to the horizontal size of the smallest atmospheric clouds, which is about one kilometer and more. At the same time pitch angle fluctuation can result in significant signal variations, if the trajectory shift caused by the angular deviation and the horizontal

characteristic scale of aerosol concentration changes are comparable.

There are many experimental observations of variations of aerosol and water vapor concentration on small vertical (about one hundred m) and horizontal (several km) scales in the lower atmosphere. Small clouds with such characteristic scales a referred to as "clusters", in order to avoid mixing them up with usual aerosol layers and clouds with the horizontal length of the order of hundred kilometers. Clusters can be produced, for example, at the final stage of the collapse of internal gravity

waves (Barenblatt and Monin, 1979) or by turbulence (Klyatskin, 2005; Klyatskin and Koshel, 2000).

Observations of Eyjafjallajokull volcano eruption in 2010 showed small cluster structures as well as huge ash clouds. In the observation carried out by Chazette et al. by Ultra-Violet Rayleigh-Mie lidar, clusters with minimal horizontal size corresponding to about 50 seconds of aircraft flight time and 80 meters thickness were found (Chazette et al., 2012, Fig.3, Fig 4). At the same time clouds with sizes up to 1 km in the vertical direction and 100 km in the horizontal direction were also observed

(Chazette et al., 2012). Layers with 1 and 2 km thickness and concentration changes about 7 times at this scale were found in

(Dacre et al., 2013, Fig.3). The same thickness with a concentration jump, which is 2 time smaller, was also found in (Turnbull et al., 2012). Simulations predict clouds with thickness about of 0.5–2 km (Hervo et al., 2012, (Fig.1)) when real observations also show thin layers with thickness of about 100 meters (Hervo et al., 2012, Fig.2, Fig.10).

Cirrus cloud split into numerous clusters with a thickness of about 100 meters at the altitudes between 6 and 11 km ((Reichardt et al., 2002),Fig.1 or (Cadet et al., 2005),Fig.2b) and stable layers with 1 km thickness ((Cadet et al., 2005),Fig.2a) were observed. Based on possible wind speed, the horizontal size of these clusters can be estimated as 3–12 km. Their concentration is changing 2–5 times in both vertical and horizontal directions at cluster scales. Ice clouds containing cluster structures with horizontal characteristic scales about hundred meters were observed, for example in ((Haarig et al., 2016), Fig.2) at altitudes about 7–11 km. Aerosol clusters in the altitude range of 1–10 km with the thickness of about 100 m and the concentration variations by 2–5 times were reported in (Burton et al., 2015, Fig.3), (Burton et al., 2014, Fig.6 dust aerosol), (Burton et al., 2015, Fig.7, Fig.13), (Burton et al., 2014, smoke aerosol in Fig.9). Clusters with the 100 meter thickness and horizontal size of about few kilometers were also observed in (Hair et al., 2008). Urban plumes measured in (Kleinman et al., 2008) also contained clusters with horizontal sizes corresponding to about of 1-2 minutes of aircraft flight time with 4 times concentration changes.

Relatively thin and long water vapor layers observed at heights below 11 km indicate a thickness of about 100 meters and more (Whiteman et al., 1992; Kiemle et al., 2008; Leblanc and McDermid, 2008). An ice layer with 100 meters vertical size can have more than 10 times concentration changes (Johnson et al., 2012).

Aerosol and water vapor clusters can be routinely observed in the atmosphere in the civil aviation flight height range. The shear of cluster with horizontal characteristic scale of about 1 km at wind speed of 20 m/s could happen in about 30–60 seconds. The evaporation and condensation effects can also influence the time of aerosol cluster evolution. On the other hand, cluster could disappear from the field of view because of pitch angle fluctuation during the same time. This creates potential ambiguity in the interpretation of the lidar backscattering signal.

In this paper, we discuss the impact of pitch angle fluctuations on both simulated and measured lidar signal in the presence of aerosol clusters with different sizes monitored by an airborne lidar. We formulate the criteria for distinguish of pitch angle fluctuation impact from the evolution of aerosol clusters. The paper is organized as follows: in Sections 2 and 3, we describe the observation model and its parameters, respectively. The simulation results are presented and discussed in Section 4. In Section 5, we make our conclusions.

## 2 Observation model and typical scales

Ground-based stationary lidar is the conventional technique for the study of the atmospheric composition, density, and aerosol properties (Zuev and Zuev, 1992). The sensing procedure is as follows: short radiation pulses are produced sequentially by a pulsed laser, each of them is transformed into a narrow beam by the optical system and sent into the atmosphere. The laser beam scatters on thermodynamic fluctuations of air density (Fabelinskii, 2012) and particles of solid or liquid aerosol (Bohren and Huffman, 2008) scatter the beam. Measured power profiles of the scattered radiation are a function of shot time $t$ and

distance $L$ to the scattering volume, the latter being derived from measured backscatter delay time $\delta t$. For a ground-based lidar with an upwards-directed beam, $L$ equals the altitude of the scattering volume and the power of the registered lidar response $I$ bears information on the atmospheric properties along the line of sight (Hauchecorne et al., 2016; Keckhut et al., 2015). As the wind drift occurs, the altitudinal cross-section of long-living aerosol clusters can be inferred from $I(L,t)$ relief images in the

$(L,t)$ plane as bars with width depending on both the wind speed and the 3D cluster structure (Haarig et al., 2016; Hoareau et al., 2012).

    The wind drift poses a significant encumbrance to studies of aerosol cluster evolution, using ground-based platforms, because it is necessary to distinguish between the temporal evolution of a particular cluster and its drift in space with the wind. While thermodynamic fluctuations of atmospheric air density in time and space may be described under the assumption of their

statistical homogeneity and stationarity, this assumption, in practice, often becomes invalid for the description of clusterized aerosol.

    For the enhancement of the civil aviation safety and flight comfort, it was suggested to use an airborne lidar with scanning the atmosphere ahead in the flight direction. The analysis of experimental results demonstrated a rapid spatiotemporal evolution of aerosol clusters (Veerman et al., 2014, Fig. 22). A schematic diagram of lidar measurements that takes account of random pitch

angle variations is shown in Fig.1. In field experiments, noise and distortions of the data are always present. One of the crucial factors is the noise related to uncontrolled fluctuations of the aircraft position and, as a result, of the airborne lidar position. In this work, we develop the results of a previous study (Gurvich and Kulikov, 2016), by the consideration of the spatiotemporal parameters of lidar images of aerosol clusters and by the assessment of the characteristic scales of clusters, at which noise caused by uncontrolled fluctuations of the aircraft position does not impede monitoring their evolution.

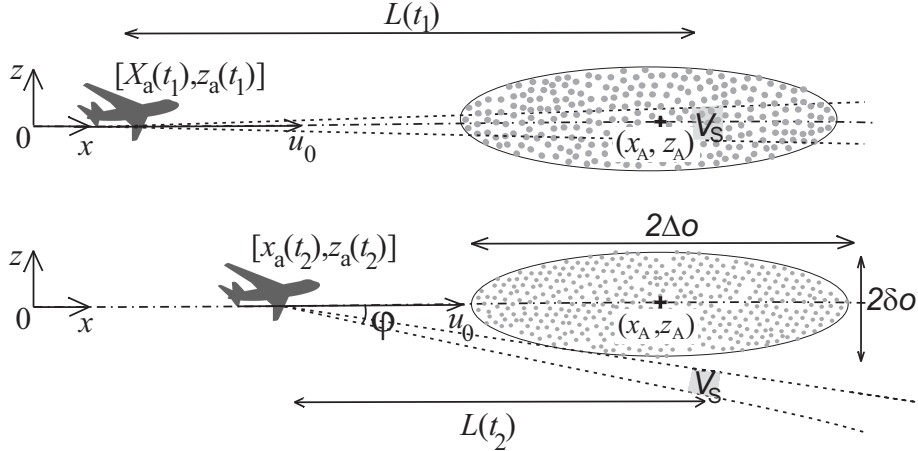

**Figure 1.** A schematic diagram of lidar measurements of the flight direction from an aircraft. The $x_a(t)$, $z_a(t)$ represent the observer's coordinates at sequential time points $t_1$ and $t_2$; the center of the observed clusters is marked with +, and their coordinates are $x_A$, $z_A$.

The fluctuations of the sensing direction during the flight can be defined by fluctuations of three angles: roll, yaw and pitch. As the horizontal size of typical aerosol formations is usually large, the azimuthal shifts of the scattering volume due to rolling

and yawing are not as significant as its vertical shift, which is characterized by the product of the observation distance $L$ and pitch angle change. For aerosol clusters with the thickness smaller or comparable to the shift of scattering volume, an incidental time modulation of the lidar response from monitored aerosol cluster may be mistaken for the cluster evolution.

Airborne lidar measurements in the flight direction suggest that it may be possible to observe evolution of the aerosol clusters with evolution time smaller than the observation time. At the same time, variations of the lidar response (Veerman et al., 2014, fig 22) could also be caused by variations of the airplane pitch. In this paper, we simulate and discuss the influence of airplane pitch angle variations on the lidar backscattered signal from the aerosol clusters.

It is evident that the backscattered signal coming from the aerosol, changes with pitch fluctuations. The scheme in Fig.1 shows that if the vertical shift of the scattering volume is $L\sin(\varphi) > \delta o$, where $L$ is the distance between the plane and the scattering volume, $\varphi$ is the angle deflection of the sensing beam from flight direction, $\delta o$ is the characteristic vertical size of the aerosol cluster, then the signal from long-living cluster contains distortions caused by scattering volume shift. These distortions may be mistaken for a result of the cluster evolution. In order to avoid the signal variations caused by the pitch angle fluctuations the condition $L\sin(\varphi)/\delta o < 1$ on the maximal acceptable beam angle deviation $\varphi$ should be fulfilled in presence of aerosol clusters with vertical size about $\delta o$.

If $L\sin(\varphi)/\delta o \geq 1$, then the aerosol cluster may occasionally disappear from the lidar's field of vision. Figure 1 is a schematic representation of the measurements with an airborne lidar that approaches a cluster (depicted by circlets) located on the flight path, with the airspeed $u_0$. The cluster thickness $2\delta o$ is much smaller than its horizontal dimension, $2\Delta o$. The flight path is shown by the dash-and-dot line. The laser beam is shown by the long-dash line. The scattering volume $V_S$, which moves with the velocity of light $c$ in measurement direction, is colored gray here. The scheme depicts two sequential time moments of measurements. In the second time moment, the beam deflects from the flight direction by angle $\varphi$ and the lidar only registers molecular scattering at the thermodynamic fluctuations of the air density.

The following typical scales of time and distance may be distinguished in the problem of lidar monitoring of the atmosphere from an aircraft in the flight direction. Assuming that the molecular scattering is negligibly weak and neglecting molecular absorption, we may accept the length of molecular extinction $L_{ext}$ to be the maximum distance. The intensity $I$ of the observed backscatter response is decreases with the distance as $L^{-2}$. Together with the sensing pulse magnitude, the internal noises of the receiver, as well as the random nature of aerosol and turbulence determine the maximum sensing distance $L_{max}$. Distance $L_{max}$ is defined as the maximal distance for which we are still able to register backscattered signal. Specifically, in our simulations we limited by signal registered with time delay corresponded to 16 km distance suppose, based on DELICAT lidar parameters, that the signal from longer distance cannot be registered due to the noise. We assume that $L_{max} < L_{ext}$. The minimum time scale is the sensing pulse duration $\tau$, which is about 10 nanoseconds for lasers used in lidars. The lengthwise dimension $l_{\parallel}$ of the scattering volume $V_S$ equals $c\tau/2$, where $c$ is the light speed. For considered pulse duration the lengthwise dimension is $l_{\parallel} = 1.5$ m. The lateral dimension $l_{\perp}$ is determined by the initial diameter $D_0$ of the sensing beam and full divergence angle $\gamma$: $l_{\perp} \simeq \gamma L + D_0$. For the typical values of $\gamma = 2 \cdot 10^{-4}$ rad, $D_0 = 10$ cm, and $L_{max} = 15$ km, the estimated value of $l_{\perp}$ is about 3.1 m at the end of the sensing path. We define the sensing path as the path during which the experimental equipment registers the backscattered lidar signal. Signal record time is determined by the pass band of the photodetector and is usually

slightly greater than $\tau$. Another characteristic time is the time interval $t_{max} = 2L_{max}/c$ of backscatter return. It determines the maximum frequency of sensing pulses. For the value of $t_{max}$ is about 0.1 milliseconds, the distance $L_{max} = 15$ km. Such a time interval is negligible compared to the time scale of detectable variations in atmospheric aerosol systems (Ivlev and Dovgalyuk, 1999). For this reason, the properties of the scattering medium, including the aerosol density and backscattering

cross-section, are considered to be invariant at time intervals $t_{max}$ when analyzing the effects of cluster evolution upon lidar images.

Lidars, in most practical cases, send recurrent pulses. In Fig. 2 they are seen as a "comb". Based on the absence of coherent relation between pulses we assume the backscattered signals to be independent for each pulse. In the hierarchy of characteristic times, the value of $t_{obs} = L_{max}/u_0$ is the time for the aircraft to approach the scatterer after the moment of its observation. The

value of $t_{obs}$ has been used in (Gurvich and Kulikov, 2016) to define long-living clusters. For observation distances from 10 to 20 km and modern aircraft velocities, this time may reach hundreds of seconds. The backscattering cross-section of aerosol particles may change significantly over the time interval of $t_{obs}$. This change is schematically depicted at Fig. 1 by the change in the number and size of scatterers.

## 3    Modeling of an aerosol cluster lidar image

For the lidar image model, we use a Cartesian coordinate system with its $Ox$ axis coinciding with the flight direction of the aircraft moving straightforward at a constant altitude. We discuss relatively small distances, $\ll \sqrt{a_E H_A}$ where $a_E$ is Earth radius, $H_A$ is atmospheric scale height. Therefore, the Earth's curvature impact can be neglected. The coordinate system origin is placed somewhere on the flight path; the $Oz$ axis is directed along the local vertical. Let's denote the aircraft position at time point $t$ as $x_a(t) = u_0 \cdot t$, $z_a(t)$.

To investigate possible artifacts generated by uncontrolled wanderings of the line of sight, which may be caused, e.g., by the fluctuations of the aircraft position, errors in the beam stabilizing system, etc., we should consider the apparent movements of the scattering volume resulting from the above factors. If the distance between the aircraft and the center of the scattering volume at time $t$ is $L$, then the coordinates $x_S$, $z_S$ of the scattering volume center are:

$$x_S(t) = x_a(t) + L \cdot cos(\varphi(t)) \cong x_a(t) + L, \; z_S(t) = z_a(t) + L \cdot \sin(\varphi(t)) \cong L \cdot \varphi(t) \tag{1}$$

Backscattered radiation is detected with the delay

$$\delta t = 2L/c \tag{2}$$

after time $t_0$ when the sensing pulse was sent. Equation (2) allows the derivation of $L$ from measured $\delta t$. Because the light velocity significantly exceeds the aircraft velocity, for the simulation purposes, it is conveniet to treat $L(t)$ and $t$, which can both be measured directly, as independent variables.

Below, we perform the analysis of the backscatter signal intensity $I(L,t)$ in the receiving aperture superimposed on the lidar output aperture. We apply the approximation of the single-scattering on aerosol particles (Ishimaru, 1978). We use the

following notations: $\rho_A(x,y,z,t)$ is the number of scatterers per volume unit, or the scatterer density, and $\sigma_{AB}(x,y,z,t)$ is the aerosol differential backscatter cross-section coefficient. For an arbitrary shaped sensing pulse with its complex envelope $U(t,t_0)$, where $t_0$ is a time moment of pulse generation, the intensity registered by the receiver at an arbitrary time point is determined by the expression (Ishimaru, 1978, Eq. 5.35):

$$I(L,t) = Cs \int_{R_1}^{R_2} \frac{\rho(R',(t-t_0)-R'/c)\sigma_B(R',t-R'/c)}{R_2}|U_i(t-R'/c,t_0)|^2 e^{-2\Gamma(R',t)}dR' \qquad (3)$$

Here $R_1 = c(t-t_0)/2$ and $R_2 = c(t-t_0+\tau)/2$ are the corresponding positions of the scattering volume boundaries, $t_0$ is the time of sensing pulse generation and $L = (R_1 + R_2)/2$ is the position of the scattering volume center along the flight route. The integration is performed along the line of sight taking into account its direction fluctuations. The factor of $exp[-2\Gamma(R,t)]$ in (3) describes the extinction and is defined by equation

$$\Gamma(R,t) = Cd \int_{x_a(t)}^{R+x_a(t)} \rho(R',t-R'/c)\sigma_T(R',t-R'/c)dR' \qquad (4)$$

Here, $\sigma_T$ is total cross-section coefficient of scattering. The product of $\rho(R',t-R'/c)\sigma_T(R',t-R'/c)dR'$ describes the total losses from molecular and aerosol scatters. Constant factors $Cs$ and $Cd$ in front of the integrals in Eqs. (3) and (4) account for the sensing pulse energy, beam geometry, receiver aperture, detector parameters etc. Equation (3) does not take into account the contribution of weak molecular scattering, which, when the measured intensity $I(L,t)$ is multiplied by $L^2 exp(2\cdot\Gamma)$, generates

a constant background on the lidar image obtained.

Because the lidar pulse is short (10 ns) in comparison to the considered spatial scales, we can use the Dirac function which significantly simplifies the analytical solution. Under this approximation, in the absence of measurement direction oscillations, signal $I(L,t)$ in receiver aperture is determined by the equation:

$$I(L,t) = I_M(L,t) + I_A(L,t)$$

$$I_M(L,t) = \frac{2C \cdot E_0}{c \cdot L^2}\rho_M(L,t-L/c)\sigma_{MB}(L,t-L/c)e^{-2\Gamma(L,t)}$$

$$I_A(L,t) = \frac{2C \cdot E_0}{c \cdot L^2}\rho_A(L,t-L/c)\sigma_{AB}(L,t-L/c)e^{-2\Gamma(L,t)} \qquad (5)$$

where the observed intensity $I$ has two components, $I_M$, resulting from the molecular scattering, and $I_A$ coming from the aerosol scattering. Here, $E_0$ is the pulse total energy, $C$ is the normalizing factor that accounts for the sensing pulse shape, the receiver aperture, detector features etc., $L$ is the distance between the lidar and the scattering volume. Equations (4) and (5) contain terms $\rho_M(R',t)\sigma_{MB}(R',t)$ and $\rho_A(R',t)\sigma_{AB}(R',t)$, which are the products of scatterers density by the cross-sections of the molecular and aerosol backscattering, respectively. The term $exp(-2\Gamma(L,t))$ describes extinction, $\rho(R',t-$

$R'/c)\sigma_T(R',t-R'/c)dR'$ represents the total losses due to molecular and aerosol extinction. This relatively simple model appears to be a good approximation for a sensing laser pulse with the duration of several nanoseconds. For the simulation purposes, we use the following normalized function for the atmospheric aerosol backscattering density:

$$\text{P(x,z,t)}= \rho_M\sigma_{MB}(x,y,z,t)/\rho_M\sigma_{MB}(max) = a\sum_q exp[-(\frac{x-x_{0q}}{\Delta o_q})^4 - (\frac{y}{\Delta y_q})^4 - (\frac{z-z_{0q}}{\delta o_q})^2 - (\frac{t-t_q}{\Delta t_q})^2] \qquad (6)$$

**Table 1.** Parameters of aerosol clusters

| Cluster | $\Delta o_q$, km | $\Delta t_q$, sec | $x_{0q}$, km | $t_q$, sec | $u_0 t_{obs}/\Delta o_q$ |
|---|---|---|---|---|---|
| A | 2.0 | 60 | 11.0 | 34 | 5.1 |
| B | 1.0 | 40 | 16.3 | 70 | 6.8 |
| C | 1.0 | 40 | 20.0 | 40 | 6.8 |
| D | 1.0 | 16 | 24.0 | 80 | 2.7 |
| E | 0.5 | 10 | 28.0 | 95 | 3.4 |

In this expression, $x$ is the axis collinear to the flight direction, $y$ is the axis perpendicular to both the flight direction and vertical axis, $z$ is the vertical axis, orthogonal to the Earth's surface below the aircraft position, $t$ is the moment of measurement, which we assume to coincide with the moment of pulse pulse generation $t_0$, due to the aforementioned smallness of the ratio $L_{max}/(c \cdot \Delta t_q) \ll 1$, $x_{0q}$, $y_{0q}$ and $z_{0q}$ are coordinates of the clusters' centers, $t_q$ is the time moment of the maximum cluster density, $\Delta t_q$ is the typical cluster evolution time, $\Delta o_q$ is the cluster scale in the flight direction, $\delta o_q$ is the typical vertical dimension of cluster. The value of $\Delta y_q$ is the transverse size of the cluster. The contribution of fluctuations of the flight direction along $y$-axis into the lidar image noise is neglegible, because the changes of scatterers' density are smooth. The parameter $\Delta y_q$ is chosen to equal $\Delta o_q$ for all the simulated clusters. The sequence of five integers q, from 1 to 5, is the sequence order of clusters along the flight path. The model parameters are summarized in the Table 1. All the five clusters have the same thickness $2\delta o_q = 2\delta o$, which was equal to $100\mathrm{m}; 300\mathrm{m}; 900\mathrm{m}$ in different simulations. Figure 2 presents the cluster sequence used in the model for $\delta o = 150\mathrm{m}$. Aerosol cluster are represented as surfaces calculated at $e^{-1}$ level of values. The distance from the initial position of the aircraft is laid off along the $Ox$ axis, the flight altitude is laid off along the $Oz$ axis and time along the $t$ axis. The aircraft velocity is assumed to be $170 \, \mathrm{m/sec}$.

The last column of the table contains the unitless ratios $u_0 \cdot t_{obs}/\Delta o_q$. Because all of them are greater than 1, we can consider our modeled clusters as long-living ones (Gurvich and Kulikov, 2016). We consider "thin" clusters, whose ratios of vertical scales to lengthwise ones are $\delta o/\Delta o \ll 1$. If such clusters are detected in the vertical direction from a ground-based platform, they are registered as layers in the altitudinal distribution of the aerosol.

Since our work is aimed at the study of the most typical features of the changes of the backscatterd lidar signal, we only discuss clusters' shape and relative size, without focusing on the type of particles that produce the signal. Consequently, the value we need to monitor is the normalized backscatter intensity $J_A(L,t) = [I_A(L,t)]/I_M(L,t)$. As the constant background coming from the scattering on density inhomogeneities does not present any interest the lidar images, all the Figures present the value of $J_A(L,t)$.

Fig. 3 shows the lidar image of aerosol clusters, modeled according to model (6). This image is simulated under the assumption of the stable flight altitude and measurement direction. In terms of Eq. (1), this means that $\varphi = 0$ and $z_S = const$, the latter value may be set to the flight altitude without restricting the generality. We focus on the problem of the impact of

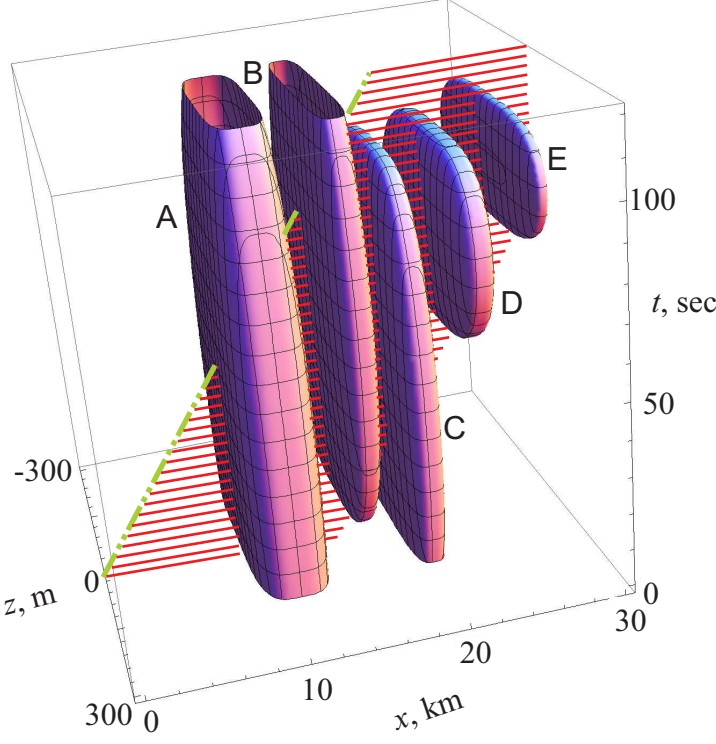

**Figure 2.** 3D images of aerosol clusters $P(x, z, t)$, calculated at $1/e$ level, for the model given by eq.(6). Dash-and-dot line is the flight trace, red "comb" represents sensing laser pulses, $L_{max} = 16$ km, $\varphi_0 = 0$.

flight parameter fluctuations upon measured lidar backscattered signal. The experiment discussed in (Veerman et al., 2014, Introduction) was conducted under clear air conditions. For this reason, for our numerical simulation, we choose the product of scatterer cross-section and density $\rho_M \sigma_{MB}$ to be equal to $2 \cdot 10^{-2}$ dB/km at the cluster's center (concentration $10^8$ particles per m$^{-3}$, and density of water $49~\mu g/m^{-3}$) (Ishimaru, 1978). This value typically corresponds to weak water aerosol clusters
5    in accordance with Fabelinskii (2012); Ishimaru (1978), which implies that the aerosol scattering does not significantly decrease the propagating laser pulse energy. The values for the other types of aerosol can be found, for example, in (Vrancken et al., 2016).

    The image $J_A$ at Fig. 3 is shown in $(L, u_0 t)$ coordinates, in which the cluster with a lifespan of $\Delta t > L_{max}/u_0$ looks like a bar, whose slope with respect to the $OL$ axis equals $\pi/4$. Longitudinal cluster scale $\Delta o$ determines the image size along the $L$
10    axis. The image size at an angle of $-\pi/4$ with respect to the $L$ axis, is determined by $u_0 \Delta t_q$ i.e. the product of aircraft speed by cluster's lifespan. Measurement of the image length $J_A$ along this direction allows, therefore, the estimation of the cluster lifespan $\Delta t_q$. If the cluster has a long lifespan, such that $u_0 \Delta t_q \gg L_{max}$, then, for a constant measurement direction, its lidar image is a homogeneous bar.

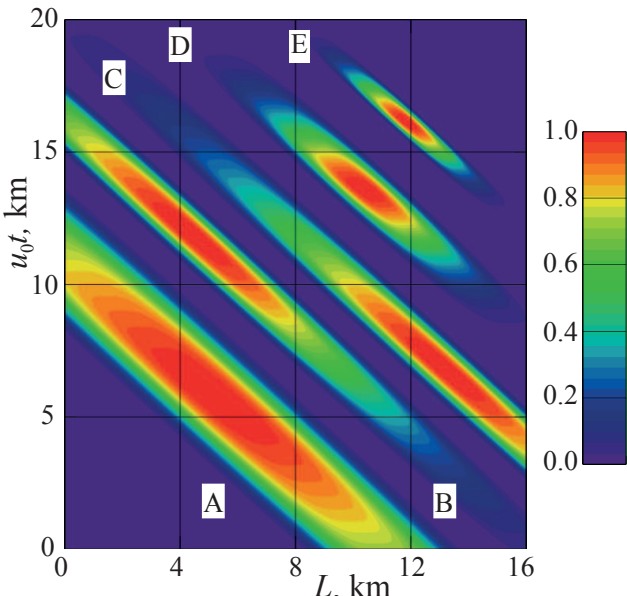

**Figure 3.** Lidar images $J_A(L,t)$ of aerosol clusters simulated according to the model (eq. (6)) for a constant beam direction aligned with the flight trace. The scale of $J_A$ is given in pseudo-color at the left. The vertical axis corresponds to the product of $u_0 t_0$ where $t_0$ is a sensing pulse generation point and $u_0$ is the aircraft speed.

## 4    The impact of measurement direction fluctuations on cluster lidar images

Under the real-world conditions, the uncontrolled variations of measurement directions always exist due to both vibrations of the carrying platform and fluctuations of flying aircraft altitude. If a cluster is strongly elongated in horizontal direction, then its lidar image is most sensitive to vertical variations of the measurement direction. For the illustration of the effects caused by

5    sensing beam deviation from the flight direction, we assume that the measurement direction, which is determined in (1) by the angle $\varphi$, changes periodically with a period of $T_\varphi = 20$ sec according to the equation

$$\varphi(t) = \varphi_0 \cdot [1 - \cos(\phi + 2\pi t/T_\varphi)] \tag{7}$$

where the normalization factor of $\varphi_0$ determines the maximum deviation angle from the flight direction, $\phi$ is the correcting parameter. Our choice of the $T_\varphi$ is based on one of the characteristic times of pitch angle fluctuations measured in the experi-

10    ment. These times vary in the range from few to tens of seconds (Fig.6 (b) and (d)). The considered effects do not disappear for smaller/larger times; such changes would result only in changing of thickness of the breaches. Given the precision character-istics of modern gyro-stabilizing devices used in civil aviation (Temp-Avia, 2016; SOMAG AG Jena, 2016), we consider here two $\varphi_0$ values: $\varphi_0 = 0.3$ and $\varphi_0 = 0.6$ degrees. We consider here $z_a(t) = 0$.

Fig. 4 shows the relation between the deviation of the scattering volume center coordinate $z_S = L \cdot \sin \varphi(t)$, the measurement

15    time $u_0 t$ and the distance $L$ between the observer and the scattering volume center. The distance $u_0 t_0 = 20$ km corresponds to

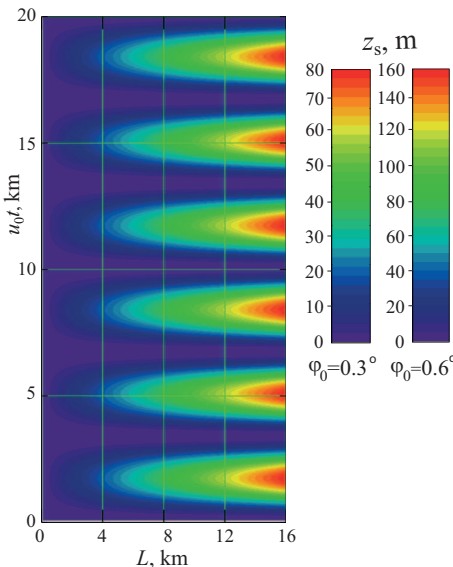

**Figure 4.** Deviation $z_S$ of the scattering volume center from the flight direction as a function of the pulse generation time $t$ and of the distance $L$.

minutes of airborne observation (for aircraft speed 170 m/sec). Beam displacement is changing from 0 to 160 m (from 0 to 80 m) for $\varphi_0 = 0.3$ degree ($\varphi_0 = 0.6$ degree) for about of 10 sec. The speed of movement was defined by period $T_\varphi = 20$ sec. The correcting parameter $\phi = 0$.

Vertical movements of the scattering volume center $z_S$ comparable to or greater than $z_S$ should be visible in the lidar image. This qualitatively follows from the description of the measurement setup in Section 2. Lidar images computed in the presence of pitch angle fluctuations in the typical range of the laser-gyros (0.1-0.2 rms) are presented in Fig. 5.

The same 5 aerosol clusters described by Eq. (6) and shown in Fig.2 are taken for lidar image simulations, but their $\delta o$ parameters that determine vertical dimensions are set to different values. Panes (a), (c) and (e) grouped in the upper row show the images simulated at lower oscillation amplitude $\varphi_0 = 0.3$ degree, for $\delta o$ values of 50, 150 and 450 m. The images at lower row panes: (b), (d) and (f), have a twice higher amplitude $\varphi_0 = 0.6$ degrees and the same values, respectively. The measurement time is 120 sec for each pane and the maximum measurement distance $L_{max} = 16$ km. Since the maximum vertical deviations of the scattering volume center coordinate $z_S$ from the flight path reach 83 m and 168 m, respectively, it is possible to consider cases with $z_S > \delta o$ and $z_S < \delta o$.

The comparison of the Fig. 3 and Fig. 5 reveals that sensing direction oscillations cause breaches in the clusters lidar images at large distances $L$ when the deviations of $z_S$ reach the maximum values. These signal fades appear due to the scattering volume shift outside cluster boundaries; the maximal shift equals $L \cdot \sin \varphi_0$. For this reason the images are more distorted at the right side of each pane of Fig. 5. Image distortions are more intense for thin clusters with low values of $\delta o$.

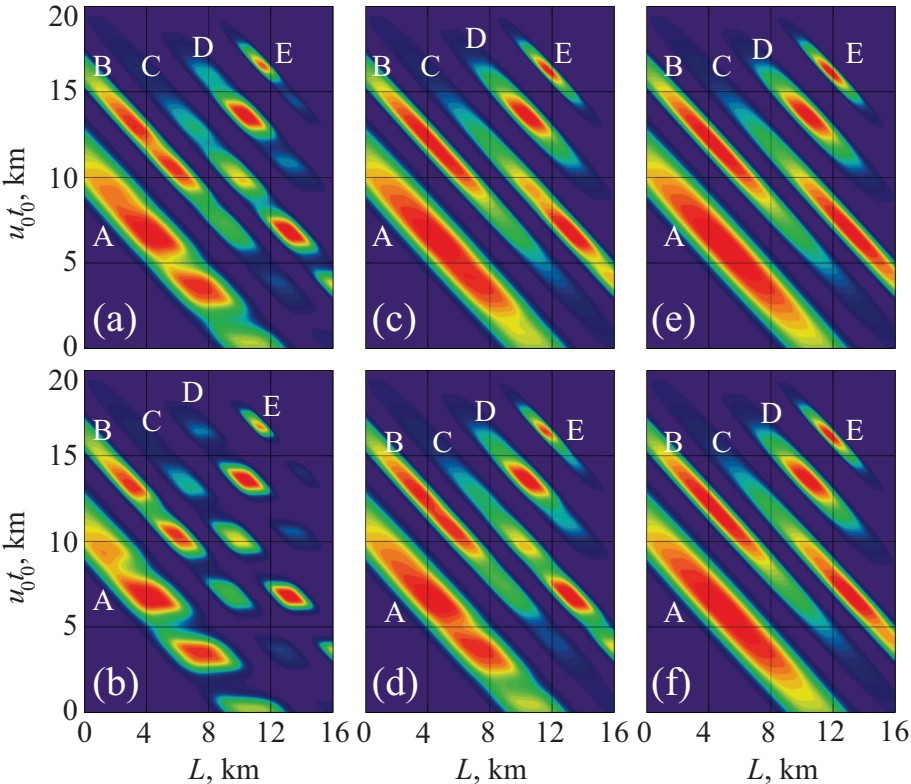

**Figure 5.** The impact of measurement direction fluctuations on the lidar image $J_A$ of aerosol clusters. The pseudo-colored scale of $J_A$ values is the same as in Fig. 3. Panes: (a), (c), (e) - oscillations amplitude $\varphi_0 = 0.3$degrees, panes: - (b), (d), (f) - $\varphi_0 = 0.6$degrees; panes: (a), (b) vertical dimension $\delta o = 50$ m, panes: (c), (d) - $\delta o = 150$ m, panes: (e), (f) - $\delta o = 450$ m.

For example, the backscattered signal at the sensing distance L=6 km at the time corresponding to aircraft trajectory coordinate $u_0 t_0 = 5$ km in presence of aerosol clusters with thickness $2\delta o = 100$ m (Fig.5ab) decreased by about 20% and 60% from the level without the pitch angle fluctuations (see Fig.3) for $\varphi_0 = 0.3$ degree and $\varphi_0 = 0.6$ degree respectively. The signal decreased by about 10% in the presence of aerosol clusters with thickness 300 m for the $\varphi_0 = 0.6$ degree and it had no noticeable changes for larger vertical sizes of cluster or smaller angles (Fig.5c,d). The backscattered signal from the aerosol layer at the sensing distance L=15 km at the time corresponded to aircraft trajectory coordinate $u_0 t_0 = 5$ km with thickness 100 m decreased by about 85% for the $\varphi_0 = 0.3$ degree and absent (only background level) for the $\varphi_0 = 0.6$ degree. The signal decrease about 35% and 45% in presence of aerosol clusters with thickness 300 m for the $\varphi_0 = 0.3$ degree and $\varphi_0 = 0.6$ degree correspondingly, while for the thickness of the cluster about 900 m the only noticeable change (about 12%) can be found for the $\varphi_0 = 0.6$ degree. Similar effects can be found in Fig.5 for each other moment of time (corresponding to flight trajectory coordinate $u_0 t_0$).

Fig.5 (a), (b) shows that the breaches appear at the same aircraft position $u_0 t_0$ for all clusters. The lines could be drawn at 2 km, 5 km (as well as at 8, 12, 15, and 18 km) in accordance with the beam direction variations. The value of $L \cdot \sin \varphi_0$ is

smaller for smaller distance $L$, consequently, the breaches "depth" is smaller for a close distance. Thus the angle $\varphi_0$ could be estimated from the intensity measurements. It may be expected that a natural process intensity, like aerosol evolution due to evaporation or condensation, varies for different clusters. A distortion due to flight direction fluctuations has the same impact on the images of all the clusters observed at the same distance.

The vertical beam deviation caused by pitch angle fluctuations is about 30 m and 60 m at 6 km distance for maximal amplitude of angle fluctuations $\varphi_0 = 0.3$ degree and 0.6 degree, respectively (Fig.4). It increases up to 75 and 150 meters for the 15 km distance. The sensing beam can easily move outside the aerosol cluster with a thickness less than the doubled shift size. Even for a movement with a smaller amplitude the backscattered signal will decrease due to decreasing of the cluster density nearby its edge.

As shown in Fig.5, the clusters with smallest evolution time corresponding to a living time below 30 sec still appeared twice for strongest fluctuations (0.6 degree) for the largest sensing distance. It means that we can observe evolution of the smallest considered cluster (0.5 km length) with the smallest considered evolution time at the considered sensing distance. The evolution of the cluster is clearly seen in decreasing the signal in the periods between the breaches caused by pith angle fluctuations. Such decreasing can be seen for all considered clusters with and without pitch angle effects (Fig.5).

For thickness values large enough, like in panes (e) and (f), the images almost do not differ from the images in Fig. 3 computed with zero $\varphi_0$ value, i.e. in the absence of measurement direction oscillations. The data presented in Fig. 5 also suggest the possibility of obtaining actual information about the vertical structure of the aerosol cluster from measurements of $\varphi(t)$ in flight.

## 5    Airborne lidar measurements in presence of pitch angle fluctuations

Laboratory of Turbulence and Wave Propagation at Obukhov Institute of Atmospheric Physics was one of the participants of the DELICAT project. We consider the results of the airborne measurements carried out in the framework of DELICAT project (Veerman et al., 2014, flight map Fig.15). The thorough analysis of CAT detection was performed in (Vrancken et al., 2016; Veerman et al., 2014; Hauchecorne et al., 2016). Here we discuss the examples of strong backscattered signal variations caused by pitch angle fluctuations which were sometimes observed during the experiments. A high-power UV Rayleigh lidar system 25 was installed on an aircraft in a forward-looking configuration as described in detail in (Vrancken et al., 2016). The DELICAT airborne lidar is based on a high-power Nd:YAG laser, which generates 7.7 ns length pulses at wavelength 1064 nm. The lidar was developed by DLR (German Aerospace Center) while the beam steering system was developed by THALES AVIONICS. The third garmonic ($\lambda = 355$ nm) with energy about 80 mJ was used for ahead sensing. The angular beam divergence was about of 200 $\mu$rad. Lidar receiver contained several subsystems such as telescope with 140 mm diameter, and optical components 30 for filtering, beam forming, stabilization, and detection. The receiver had two channels: for co- and cross-polarization. Lidar range resolution was about 5 m. Further details of the experimental setup can be found in (Veerman et al., 2014; Vrancken et al., 2016).

The turbulence area detection was based on the lidar measurements of the fluctuation in density of air associated with the turbulent wind (Feneyrou, 2009; Vrancken et al., 2016; Hauchecorne et al., 2016). This idea was tested at first with using of the ground-based lidar (Hauchecorne et al., 2016). Detail discussion of the $C_n^2$ evaluation method and experimental examples of turbulence lidar signal responses with estimated values of $C_n^2$ can be found, for example, in the Chapter 4b of the Ref.

(Hauchecorne et al., 2016) or in (Vrancken et al., 2016).

In the Fig.6 only co-polarized component is shown. For the case that we discuss below, it only differs from the cross-polarized component by the amlitude coefficient. The measured intensity is normalized in order to compensate the signal decay with the distance $I(L,t)_{norm} = I(L,t) * (R/R_{2km})^2$ and presented in Fig.6. Though the flight routes for the DELICAT experiments were chosen in order to avoid large amount of aerosol, the signal variations caused by aerosol backscattering was significant

(Fig.6 (b) and (d)). The civil aviation routes can include more aerosol clouds.

We only present a few minutes of flight N9 measured in France, August 8, 2013. The measurements presented in Fig.6a were acquired during the time interval from 20.22 to 20.23 UTC time, between the geographical positions (46.26,6.38) and (46.33,6.48) at the altitude of 9.46 km. The measurements presented in Fig.6d were acquired during the time interval from 20.32 pm to 20.33 pm UTC time, between the geographical lattitude/longitude positions (47.20,6.49) and (47.31,6.49) at a

altitude of 10 km. The aircraft speed was about of 170m/s in both cases. The backsckattered signal contains noise caused by different sources. The lidar signal correction from molecular attenuation is presented in (Veerman et al., 2014, Fig.17). It is mentioned there that the lidar signal is exploitable from 3 km to 15 km due to saturation effect. In order to avoid this problem completely and be sure that noises due to equipment instability do not impact on our research results we chose 4 km as minimal distance for signal analysis.

The experiment shows that the yaw as well as roll angle fluctuations did not exceed the pitch angles ones (excluding few moments of significant elevations/descent moments during the flight). The altitude changes during the flight, excluding few areas of the significant elevations/descent did not exceed a value of about ten meters. In accordance with the sensing geometry under discussion and possible sizes of the aerosol clusters, only pitch angle fluctuations can result in noticeable signal changes. The pitch angle fluctuations presented in Fig.6b corresponded to the lidar backscattered signal presented in Fig.6a;

both bacscattered signal and pitch angle fluctuations for the other observation interval are shown in Fig.6d. One can see that backscattered signal breaches appeared simultaneously with the pitch angle fluctuations.

The experimental observations shown in Fig.6 (b) and (d) demonstrate that there are fast and slow pitch angle fluctuations, with the characteristic time scales of 3-4 sec and about 10-20 sec, respectively. The dotted lines in panels (a), (b) and (d) highlight period of these fluctuations. For the visual convenience only periods of few fast fluctuations in Fig.6 (b) and only few

slow fluctuations in Fig.6d are highlighted. The pitch angle fluctuations result in significant changes of backscattered signal. This impact can be seen, for example, in Fig.6a where each signal breach is a result of corresponded pitch angle fluctuations. Two significant signal changes due to slow pitch angle fluctuations can be seen in Fig.6d. The two clusters, first at the $u_0 t_0 = 5$ km (30 sec) and second at $u_0 t_0 = 8$ km (50 sec), are suddenly appeared in the field of view due to significant change of the pitch angle presented in the figure by the red curve. In order to resolve the features of backscattered signal caused by the slow

pitch angle fluctuations, this type of fluctuations was chosen for the numerical simulation section.

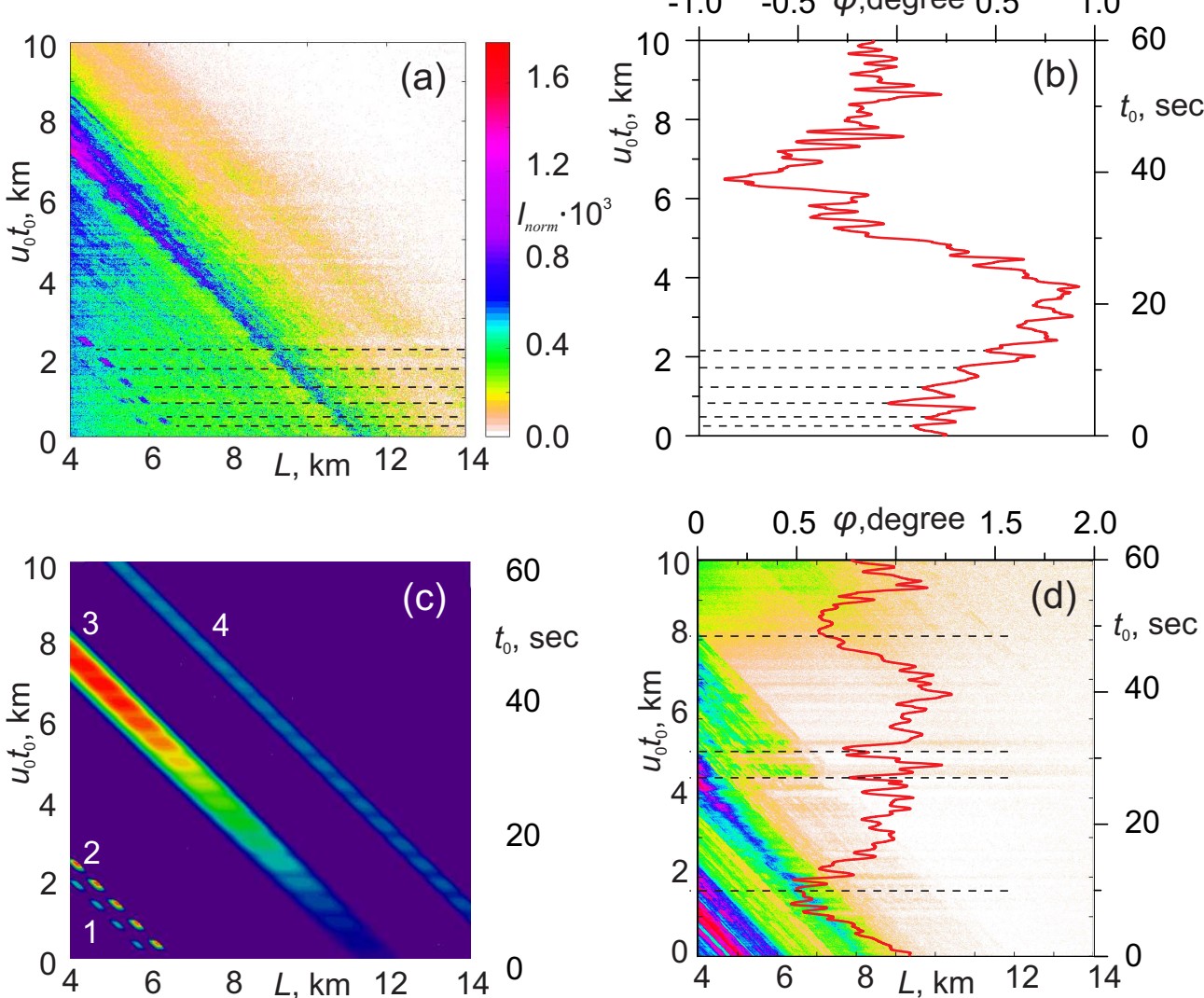

**Figure 6.** The experimental data and numerical simulations: (a) the normalized intensity $I_{norm}$ measured during 1 minute at airborne experiments (20.22-20.23); (b) measured pitch angle fluctuations correspond to (a); (c) simulations of the exeriment presented in (a), the clusters are marked by their numbers; (d) the normalized intensity $I_{norm}$ measured during 1 minute at airborne experiments (20.32-20.33).

Consider the first and second clusters in Fig.6a which firstly detected at distances 6 km and 14 km, respectively. It can be seen that there are breaches in the signal which appeared simultaneously in both responds. The value of observed signal was decreased by 3 times from the undisturbed value in the breaches. The breaches demonstrate the same behavior as simulated (see Fig.6c) which is typical for the case of presence both uncompensated pitch angle fluctuations and aerosol clusters.

In order to simulate observed in Fig.6a effect, we chose four clusters with parameters presented in Table 2. The results of simulations are presented in Fig.6c. The density of the first and last cluster was chosen to be twice less than for the second

**Table 2.** Parameters of aerosol clusters: the simulation of the experiment

| Cluster | $\Delta o_q$, m | $\Delta t_q$, sec | $x_{0q}$,km | $t_q$,sec | $\delta o_q$, m |
|---------|------|------|------|------|------|
| 1 | 100 | 120 | 5.7 | 34 | 50 |
| 2 | 100 | 120 | 6.2 | 40 | 50 |
| 3 | 600 | 26 | 11.1 | 40 | 500 |
| 4 | 300 | 120 | 20.0 | 53 | 500 |

and third cluster. The first and the second clusters have vertical thickness $2\delta o$ about 100 meters, while the third and fourth one have thickness about 1000 meters. For this reason, the pitch angle effect on the variations of the backscattered signal from the last clusters is weak. The period of pitch angle fluctuations was chosen $T_\varphi = 2.85$ sec in accordance with observed fast fluctuations (Fig.6a). The correcting parameter $\phi = 0.5\pi$. The maximal amplitude $\varphi_0$ of the pitch angle fluctuations in the simulation was 0.6 degree (corresponds to 0.2 degree rms). This parameters of the clusters and pitch angle fluctuations allows to fit the experimentally observed decreasing of the signal level and time interval between the signal reappearance (sizes of the breaches). Based on our numerical analysis we can conclude that the characteristic vertical size of the aerosol clusters provide a noticeable impact on the backscattered signal is about 50-100 meters. The decreasing of vertical size of the aerosol clusters would increase of this estimation.

If we assume that $\varphi_0 = 0.15$ degree (corresponds to 0.05 degree rms),the experimental results can be approximated with the cluster thickness 25-30 m.

## 6   Conclusions

In this paper the influence of fluctuations of the flight parameters upon images acquired by an airborne lidar system sensing ahead of the aircraft along the flight direction have been discussed in regard to the dependence on characteristic sizes of aerosol layers. It is shown that the pitch angle fluctuations are the important parameter for the airborne lidar sensing ahead the flight direction in the case when their uncompensated value result in the sensing beam shift about the vertical size of the aerosol clusters. We performed numerical simulations, which demonstrate the pitch angle fluctuation impact upon lidar signal. The simulations cover the thicknesses of atmospheric aerosol clusters in the range of tens and thousand of meters accounting for realistic values of pitch angle fluctuations. We also show that LIDAR sensing ahead along the flight direction can potentially provide information about aerosol temporal evolution characteristics even in presence of pitch angle fluctuations for reasonable cluster size and evolution time at the considered sensing distance.

We demonstrate that pitch angle fluctuations can have a noticeable impact upon measurements of the backscattered signal, even for a lidar with the system of compensation of the angle fluctuations. Numerical simulations predict that uncontrolled fluctuations can result in signal noise including extreme fades and spikes. We show that the aerosol concentration variations

on a scale of 100-300 m have a significant impact on the backscattered signal, if the correction for the angular fluctuation has a residual rms error about of 0.1–0.2 degrees, which is typical for beam steering systems used in the civil aviation. Fluctuation influence is shown to depend on the characteristic vertical size of atmospheric aerosol clusters and to introduce larger errors for aerosol density variations on smaller vertical scales. We formulate criteria for distinguishing this impact from the temporal

evolution of atmospheric aerosol clouds.

The lidar backscattered signal from 15 km sensing distance can disappear (or decrease by about 85%) for compensation of pitch angle fluctuations with 0.2 degree rms (0.1 degree rms) in presence of aerosol clusters with characteristic vertical scale about 100 meters. Aerosol clusters with thickness about 300 meters lead to 45% (35%) signal decreasing for the same sensing distance and pitch angle fluctuations. The signal level fluctuations about 60% (20%) can be caused by pitch angle

fluctuations with 0.2 degree rms (0.1 degree rms) at the 5 km sensing distance. Pitch angle fluctuations in presence of aerosol clusters with thickness about [100-300] m with angular correction about [0.1-0.2] degree rms lead to noticiable breaches in the backscattered signal. Presence of two or more aerosol clusters allows to easily distinguish the areas of significant beam wander due to signal decreasing caused by pitch angle fluctuations.

We presented and discussed an example of airborne lidar experimental observations from the DELICAT project that shows

signal variations simultaneously appearing from different aerosol clusters consistent with the signal fades caused by the impact of pitch angle fluctuations in accordance with measurements of the pitch angle fluctuations. Simulations of the experiment are performed with assuming the aerosol clusters thickness about 100 meters (1000 m for the large cluster) for the case of pitch angle compensation with 0.2 degree rms. For compensation with 0.05 degree rms noise the corresponded value of the aerosol clusters' thickness 25-30 meters (about 250 meters for the large cluster).

The signal from the areas with significant pitch angle fluctuations can be used only with additional assumptions due to the fact that the sensing beam deviates from the flight trajectory. We need to assume that turbulence strength does not significantly change at the scale of this deviation which is fulfilled only for the short distances and small angle fluctuations. Otherwise this deviation would lead to the turbulence strength estimation changes which cannot be corrected due to absence of backscattered signal from the actual aircraft trajectory. At the same time, generally speaking, the aerosol clusters' evolution in absence

of significant uncompensated fluctuations of the pitch angle should not prevent to the turbulence strength estimation. The numerical simulations shown that, for reasonable parameter range, these cases can be distinguished.

*Acknowledgements.* The performed analysis was based on the measurements performed by lidar which was developed by DLR (German Aerospace Center) with the beam steering system developed by THALES AVIONICS SA. The authors grateful to the colluagies from DELICAT project for the experimental data and helpful questions. The authors are grateful to M.E. Gorbunov and O.V. Fedorova for the thorough

manuscript review, A.V. Shmakov for fruitful discussions, and F. Dalaudier for turning our attention to the significance of measurement direction control. Work on Sections 1-3 was supported by the Russian Science Foundation (grant RSCF No. 14-27-00134). Work on Sections 4 and 5 was supported by Russian Foundation for Basic Research (grant No. 16-05-00358-a).

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
