# Peer review of "Impact of pitch angle fluctuations on airborne lidar sensing ahead along the flight direction"

_Atmospheric Measurement Techniques, 2017_

## Referee Comment (RC1) · V. Banakh (Referee) · 28 Jun 2017

The paper is devoted to analysis of the variations of airborne lidar echo signal power when probing the atmosphere ahead along the flight direction to detect the areas of high intensity of the turbulence (clear air turbulence (CAT)). Starting from the detailed review of possible sources of the errors, the authors conclude that uncompensated pitch angle variations in the presence of aerosol thin clusters can lead to noticeable echo signal power changes which can be mistakenly interpreted as the impact of turbulence or fast variations of aerosol concentration. Based on the performed analysis the authors formulate the criteria for distinguishing the pitch effect from the temporal

evolution of atmospheric aerosol concentration. The obtained theoretical and computer simulation results are applied to interpretation of the airborne lidar experimental observations. There are few notations to the manuscript material. 1. As the authors formulate, the reason for the considered in the paper task is revealing the possible errors in airborne lidar detection of CAT. At the same time there is no information in the manuscript which method is used for recognition of the CAT areas. There is no analysis of expected variations of lidar echo signal power caused by CAT in comparison with the variations of that because of the pitch effect. It is not clear what is comparative contribution of the CAT and pitch effect to the total echo signal power variations. 2. Strong inhomogeneity of aerosol concentration is serious problem in interpretation of results of lidar remote sensing the turbulent atmosphere. To exclude the uncertainty in lidar determination of intensity of turbulence caused by variations of aerosol concentration along probing path, two equivalent receiver channels are used [1-5], for example. Some comment on possibility of application of similar approach to avoid impact of pitch effect in airborne lidar detection of CAT may be useful in the paper. 3. It is known (see works by A.S. Gurvich) that at the heights of about 10 km and above the refractive turbulence is strongly anisotropic one and turbulent inhomogeneities have vertical dimensions much less than horizontal ones similar to thin aerosol clusters considered in the paper. These inhomigeneities can cause the refraction of probing beam. Estimation of impact of atmospheric optical refraction on probing beam propagation direction as compared to the pitch angle variations may be useful. 4. There is very detailed introduction in the manuscript which contains a lot of information in the paper subject. But part of them is not necessary. For example, it is obviously that nonlinear effects (filamentation) can not be expected for probing nano pulses with pulse energy about hundred of mJs used in typical lidars. Conclusion: the manuscript contains very useful results which can find the application in interpretation of the experiments in airborne lidar probing the atmosphere. The manuscript can be published in the AMT with taking into account the listed notations above. 1. Banakh V.A., Razenkov I.A., Smalikho I.N. Aerosol lidar for study of the backscatter amplification in the atmosphere. Part I.

Computer simulation. // Optika Atmosfery i Okeana. 2015. V. 28. No. 01. P. 5-11 [in Russian]. 2. Banakh V.A., Razenkov I.A. Aerosol lidar for study of the backscatter amplification in the atmosphere. Part II. Construction and experiment. // Optika Atmosfery i Okeana. 2015. V.28. No. 02. P. 113-119 [in Russian]. 3. Banakh V.A., Razenkov I.A., Smalikho I.N. Laser echo signal amplification in a turbulent atmosphere // Applied Optics. 2015. Vol.54. No. 24. P. 7301-7307. 4. Banakh V.A., A.; Razenkov, I. A. Lidar measurements of atmospheric backscattering amplification // Optics and Spectroscopy. 2016. V.120. No.2. P.326-334. 5. Banakh V.A., Razenkov I.A. Refractive turbulence strength estimation based on the laser echo signal amplification effect // Optics Letters. 2016. V. 41. No.19. P.4429-4432.

---

## Referee Comment (RC2) · Anonymous Referee #2 · 7 Jul 2017

General Comments: This study deals with the impact of the pitch angle fluctuations on airborne lidar sensing along flight direction. The authors formulated the criteria that allow the recognition of signal variation and provided an example from the onboard lidar data obtained during a DELICAT framework, indicating that a noticeable pitch angle fluctuation impact is presented. The manuscript is well written and can be published in Atmospheric Measurement Techniques journal, however I would suggest to the authors to take the following comments in to consideration. Minor Comments: 1. The abstract provides a detailed information however it would be nice to provide also some values from their results. 2. Page 4, line24: Please consider providing reference at the end of this line. 3. Page 13, line 15: The authors are kindly requested to define

if it is full or half angle the corresponding angular beam value that is provided in the manuscript. 4. Page 13, lines 26: I would kindly suggest to the authors to provide the latitude/longitude points with less decimal numbers if possible. Please correct this through the entire manuscript. 5. Page 13, line 30: The authors mention that they choose the spatial window 4-14 km for their experimental study, because it is almost free from other noise factors. Please comment more on this decision. 6. One line later at the same page (Line 31) they state that the data from Figure 6a are provided with no pitch angle fluctuation and in Figure 6b with pitch angle fluctuation. The authors are kindly suggested to state (in the manuscript and in the corresponding caption of Figure 6) much this fluctuation is. 7. Figure 6: For reasons of clarity it would be useful for the reader if the color bars of this figure were within the same limits. I would kindly suggest to the authors to provide the z axis with the same min max values and interval.

---

## Author Comment (AC1) · 2 Aug 2017

Dear Dr. Banakh,

Thank you very much for your positive feedback and constructive suggestions. I appreciate your remarks and am glad to provide more information on each of your comments. This information will be included in the paper.

1. "As the authors formulate, the reason for the considered in the paper task is revealing the possible errors in airborne lidar detection of CAT. At the same time there is no information in the manuscript which method is used for recognition of the CAT areas.

[Figure]

There is no analysis of expected variations of lidar echo signal power caused by CAT in comparison with the variations of that because of the pitch effect. It is not clear what is comparative contribution of the CAT and pitch effect to the total echo signal power variations."

The thorough analysis of CAT detection by DELICAT lidar is given in References (Vranchken et al., 2016; Veerman et al., 2014; Hauchecorne at el., 2016). In a few words, a high-power UV Rayleigh lidar system was installed on an aircraft in a forward-looking configuration as described in detail in (Vranchken et al., 2016). The turbulence area detection was based on the lidar measurements of the fluctuation in density of air associated with the turbulent wind (Feneyrou et al., 2009; Vranchken et al., 2016; Hauchecorne at el., 2016). This idea was tested at first with using of the ground-based lidar (Hauchecorne at el., 2016). Detail discussion of the $C_n^2$ evaluation method and experimental examples of turbulence lidar signal responses with estimated values of $C_n^2$ can be found, for example, in the Chapter 4b of the Ref. (Hauchecorne at el., 2016).

Comparison lidar echo signal caused by CAT and signal variations caused by pitch angle fluctuations should account the fact that pitch angle fluctuations can lead to both turbulence and aerosol signals. When the sensing beam strays from the forward prop-agation and goes to the area with different turbulence the lidar echo signal is chang-ing proportionally to ratio between both turbulence on the flight trajectory and turbu-lence which the strayed beam sense. Similarly, the contribution of aerosol response in the presence of pitch angle fluctuations depends on comparable aerosol density (or backscattering coefficient) in forward and strayed directions. The signal variations due to pitch angle fluctuations can fall down to the background level as presented in simulations and experiment Fig.5(b,f), Fig.6b. The aerosol lidar signal observed in the experiment was comparable with turbulence strength $C_n^2 = 2.5 \cdot 10^{-16} m^{-2/3}$. Estima-tions of expected turbulence signal respond based on the assumptions of the value of the structure characteristic $C_n^2 = 2.5 \cdot 10^{-16} m^{-2/3}$ were performed by Dr. Vorobiev in the framework of the DELICAT project.

Thus, the pitch angle fluctuations can lead to signal level changes from the background level up to the level of response of the turbulence/aerosol which is present in the area sensed by the strayed beam.

References 1

Feneyrou, P., Lehureau J.-C., and Barny H.: Performance evaluation for long-range turbulence-detection using ultraviolet lidar, Applied Optics, 48, 3750–3759, 2009.

Veerman, H. P. J., Vrancken, P., Lombard, L.: Flight testing delicat - a promise for medium-range clear air turbulence protection, European 46th SETP and 25th SFTE Symposium 2014, Lulea, Sweden, 2014.

Vrancken, P., Wirth, M., Ehret, G., Barny, H., Rondeau, P., Veerman, H.: Airborne forward-pointing UV Rayleigh lidar for remote clear air turbulence detection: system design and performance. Applied optics, 55 (32), 9314-9328 2016.

A. Hauchecorne, C. Cot, F. Dalaudier, J. Porteneuve, T. Gaudo, R. Wilson, C. Cenac, C. Laqui, P. Keckhut, J.-M. Perrin, A. Dolfi, N. Cezard, L. Lombard, and C. Besson: Tentative detection of clearair turbulence using a ground-based Rayleigh lidar, Applied Optics 55, 3420–3428, 2016.

2. "Strong inhomogeneity of aerosol concentration is serious problem in interpretation of results of lidar remote sensing the turbulent atmosphere. To exclude the uncertainty in lidar determination of intensity of turbulence caused by variations of aerosol concentration along probing path, two equivalent receiver channels are used [1-5], for example. Some comment on possibility of application of similar approach to avoid impact of pitch effect in airborne lidar detection of CAT may be useful in the paper."

The two channel scheme based on backscattering enhancement (BSE) looks promising for future airborne applications in light of both thorough theoretical analysis and experimental evidence of success reported in (Banakh and Smalikho, 2011; Banakh et al., 2015; Banakh and Razenkov, 2016a; Banakh and Razenkov, 2016b).

The BSE effect for the optical waves which encounter an obstacle in turbulent medium was initially found in the theoretical research (Vinogradov et al., 1973) and then experimentally confirmed (Gurvich and Kashkarov, 1977). In framework of DELICAT project the idea of possible turbulence strength estimation based on BSE was theoretically analyzed and reported (Gurvich 2012; Gurvich and Kulikov 2013).

References 2

Banakh, V. A. and Smalikho, I. N.: Determination of optical turbulence intensity by atmospheric backscattering of laser radiation, Atmos. Oceanic Optics, 24 (5), 457–465, 2011.

Banakh V.A., Razenkov I.A., Smalikho I.N.: Laser echo signal amplification in a turbulent atmosphere, Applied Optics, 54 (7), 7301-7307, 2015.

Banakh V.A., A.; Razenkov I. A.: Lidar measurements of atmospheric backscattering amplification, Optics and Spectroscopy, 120 (2), 326-334, 2016.

Banakh V.A., Razenkov I.A.: Refractive turbulence strength estimation based on the laser echo signal amplification effect, Optics Letters, 41 (19), 4429-4432, 2016.

Vinogradov, A. G., Kravtsov, Y. A. and Tatarskii, V. I.: Backscattering enhancement effect on bodies placed in a medium with random inhomogeneities, Izv. Vyssh. Uchebn. Zaved., Radiofizika 16 (7), 1064–1070, 1973.

Gurvich, A. S. and Kashkarov, S. S.: On the issue of scattering enhancement in a turbulent medium, Izv. Vyssh. Uchebn. Zaved., Radiofizika 20 (5), 794–800, 1977.

Gurvich, A. S.: Lidar sounding of turbulence based on the backscatter enhancement effect, Atmospheric and Oceanic Physics, 48 (6), 585–594, 2012.

Gurvich, A. S., and Kulikov V. A.: Lidar sensing of the turbulence based on the backscattering enhancement effect, SPIE LASE, Free-Space Laser Communication and Atmospheric Propagation XXV, pp. 86100U-86100U, International Society for Optics and Photonics, 2013.

3. "It is known (see works by A.S. Gurvich) that at the heights of about 10 km and above the refractive turbulence is strongly anisotropic one and turbulent inhomogeneities have vertical dimensions much less than horizontal ones similar to thin aerosol clusters considered in the paper. These inhomigeneities can cause the refraction of probing beam. Estimation of impact of atmospheric optical refraction on probing beam propagation direction as compared to the pitch angle variations may be useful."

The papers devoted to theoretical and experimental research of the atmospheric anisotropy (Gurvich, 1984; Gurvich, 1997; Gurvich and Brekhovskikh, 2001; Gurvich and Kan, 2003a,b; Gurvich and Chunchuzov, 2003; Sofieva at el 2010) contains consideration of long paths about few hundreds of kilometers. In these research papers the signal transmittance from satellite to satellite or observations of star scintillations from the satellite-borne sensor through the atmosphere were considered. The turbulence anisotropy can noticeable bend the light propagated over such long distances, but this impact is almost negligible for short fifteen km optical path. Possible laser beam trajectory deviation of about ten meters is small taking into account the thickness of cluster discussed in our paper (100 meters).

At the same time, refractive layers can also significantly change the trajectory of optical wave propagation (der Werf, 2003; Nunalee 2015). The consideration of such effects can be performed in the framework of geometrical (Southwell, 1982; Werf, 2003; Nunalee, 2015) or wave optics (Vorontsov and Kulikov 2015, Kulikov et al, 2017). Both turbulence anisotropy and possible impact of refractive layers should be considered in the case of extended sensing distances.

References 3

Gurvich, A. S.: Fluctuations in the observations of extraterrestrial cosmic sources through the earth's atmosphere, Radiophysics and Quantum Electronics, 27(8), 665-672, 1984.

Gurvich, A. S.: A heuristic model of three-dimensional spectra of temperature inhomogeneities in the stably stratified atmosphere, Annales Geophysicae, 15(7), Springer Berlin/Heidelberg, 1997.

Gurvich, A. S., and Brekhovskikh V. L., Study of the turbulence and inner waves in the stratosphere based on the observations of stellar scintillations from space: A model of scintillation spectra, Waves in Random Media 11(3), 163-181, 2001.

Gurvich, A. S., and Kan V.: Structure of air density irregularities in the stratosphere from spacecraft observations of stellar scintillation: 1. Three-dimensional spectrum model and recovery of its parameters, Izvestiya - Atmospheric and Oceanic Physics, 39(3), 300-310, 2003.

Gurvich, A. S., and Kan V.: Structure of air density irregularities in the stratosphere from spacecraft observations of stellar scintillation: 2. Characteristic scales, structure characteristics, and kinetic energy dissipation, Izvestiya - Atmospheric and Oceanic Physics, 39(3), 311-321, 2003.

Gurvich, A. S., and Chunchuzov I. P.: Parameters of the fine density structure in the stratosphere obtained from spacecraft observations of stellar scintillations, Journal of Geophysical Research: Atmospheres, 108.D5, 2003.

Sofieva, V. F., Gurvich A. S., and Dalaudier F.: Mapping gravity waves and turbulence in the stratosphere using satellite measurements of stellar scintillation, Physica Scripta, T142, 014043, 2010.

Vorontsov, M. A., and Kulikov, V. A.: Framework for analysis of joint impact of atmospheric turbulence and refractivity on laser beam propagation, OSA conference: Propagation through and Characterization of Distributed Volume Turbulence and Atmospheric Phenomena pp. PM4C-1, 2015.

Kulikov, V. A., Basu, S., and Vorontsov, M. A.: Simulation of laser beam propagation based on mesoscale modeling of optical turbulence and refractivity, OSA conference:

[Figure]

Propagation Through and Characterization of Atmospheric and Oceanic Phenomena, PTh3D-3, 2017.

Southwell W. H., Ray tracing in gradient-index media, J. Opt. Soc. Am. 72, 908-911, 1982.

der Werf S. Y., Ray tracing and refraction in the modified US1976 atmosphere, Appl. Opt. 42, 354-366, 2003.

Nunalee C. G., He P., Basu, S., Minet J., and Vorontsov M. A.: Mapping optical ray trajectories through island wake vortices, Meteorology and Atmospheric Physics, 127(3), 355-368, 2015.

4. "There is very detailed introduction in the manuscript which contains a lot of information in the paper subject. But part of them is not necessary. For example, it is obviously that nonlinear effects (filamentation) can not be expected for probing nano pulses with pulse energy about hundred of mJs used in typical lidars."

That is true that non-liner effects should not occur during propagation of the laser pulses emitted by the DELICAT lidar. The discussion of non-linear effects is included in Introduction because of the possible future implementations which may demand more accuracy or longer propagation distance, and therefore demand shorter pulses with higher energy.
* * *

---

## Author Comment (AC2) · 2 Aug 2017

Dear anonymous Referee,

Thank you very much for your comments. I appreciate your suggestions and am glad to update the manuscript following your comments.

1. "The abstract provides a detailed information however it would be nice to provide also some values from their results."

The pitch angle fluctuations uncompensated by gyro-platform with residual rms error about 0.1–0.2 degrees in presence of aerosol concentration variations on a scale of

100-300 m can have a significant impact on the level of backscattered signal changing it a few times or even more.

2. "Page 4, line24: Please consider providing reference at the end of this line."

The analysis of experimental results demonstrated a rapid spatiotemporal evolution of aerosol clusters (Veerman et al., 2014, Fig. 22).

3. "Page 13, line 15: The authors are kindly requested to define if it is full or half angle the corresponding angular beam value that is provided in the manuscript."

The full angular beam divergence was about of 200 rad.

4. "Page 13, lines 26: I would kindly suggest to the authors to provide the latitude/longitude points with less decimal numbers if possible. Please correct this through the entire manuscript."

The measurements presented in Fig.6a were acquired during the time interval from 8.32 pm to 8.33 pm UTC time, between the geographical lattitude/longitude positions (47.20,6.48) and (47.30,6.48) at a height of 10km. The measurements presented in Fig.6b were acquired during the time interval from 8.22 pm to 8.23 pm UTC time, between the geographical positions (46.26,6.37) and (46.32,6.48) at a height of 9.46 km.

5. "Page 13, line 30: The authors mention that they choose the spatial window 4-14 km for their experimental study, because it is almost free from other noise factors. Please comment more on this decision."

The lidar signal correction from molecular attenuation is presented in Fig.17 (Veerman et al., 2014). It mentioned there that the lidar signal is exploitable from 3 km to 15 km due to saturation effect. In order to avoid this problem completely and be sure that noises due to equipment instability do not impact on our research results we chose 4 km as minimal distance for signal analysis.

6. "One line later at the same page (Line 31) they state that the data from Figure 6a are provided with no pitch angle fluctuation and in Figure 6b with pitch angle fluctuation. The authors are kindly suggested to state (in the manuscript and in the corresponding caption of Figure 6) much this fluctuation is."

At least three aerosol cluster's responses can be found at the Fig.6b. The first two clusters, firstly detected at distances 6 km and 6.5 km, respectively, are weak. It can be seen that there are breaches in the signal which appeared simultaneously in both responds. The breaches demonstrate the same behavior as simulated (see Fig.5) for the case of presence of uncompensated pitch angle fluctuations. The value of observed signal was changed about 3 times.

7. "Figure 6: For reasons of clarity it would be useful for the reader if the color bars of this figure were within the same limits. I would kindly suggest to the authors to provide the z axis with the same min max values and interval."

Thank you for the suggestion. The limits of color bars are changed.
* * *
[Figure]

[Figure]

(a)                                    (b)

**Fig. 1.**

---

## Short Comment (SC1) · 4 Aug 2017

General: This paper aims at comparing airborne horizontal lidar measurements of aerosol clusters (or possibly hydrometeor clusters / small clouds) is the presence of beam angle fluctuations obtained during an airborne campaign with numerically modelled lidar signals for the same case. The declared purpose (in the abstract) is to give a boundary condition in order to separate lidar signal variations attributed to turbulence (molecular density variations) from variations due to beam angle fluctuations. However, the suite of the abstract and the whole paper concentrates on the effect of such beam movement on the observation of (hypothetic) aerosol clusters with limited vertical extent (such as small Cirrus). Such is also the declared goal of line 10 of the abstract to "formulate criteria" for identifying signal variations resulting from beam angles deviations probing such clusters. However, the whole paper does not deliver such criteria. It gives long introductions into cloud clusters, the measurement geometry and lidar signal analysis. Then, when comparing simulated lidar signals with real-world measurements, the paper falls short of expectations: - The assumptions taken on values of the beam movement are rather arbitrary and not well-founded as are the parameters of the aerosol "clusters" - Regarding the real airborne measurements, the flight data (from the IS, notably) are not taken into account for reference. - The individual analysis and the respective comparison are performed rather qualitatively and "by eye", I cite here: page 13, lines 3-4 – "more distorted", "more intense". A quantitative statement is lacking here. - The respective chapters 4 and 5 are very short in content and discussion. - The conclusion effectively is none, because there is no quantitative measure that has been determined. The only conclusion that is valid is that such pitch variations of the sensing beam does generate fluctuations in the (normalized) signal when encountering a vertically extended object. But this is geometrically obvious. - The promised conclusion on separation of turbulence signal fluctuation (which is on the sub-percent level, Vrancken et al. 2016) from pitch/cluster variation is not given since mere arbitrary color bars are given without any notion on temporal characteristics. In addition to above described contentual shortcomings, the text also yields many conceptual mishaps, some of them described here below. Some of them may be founded in language problems, other in hastily written assertions (page 6, lines 25ff, e.g.). In summary, I suggest at least a major revision of this document with not only incorporating the points listed here, but also delivering resilient numbers. This should include a thourough study and analysis of a whole range of possible pitch variations (amplitude/time constant-bandwidth), cluster vertical extent, cluster scattering/concentration distribution and temporal evolution of the latter. This was promised in the abstract and introduction. Regarding turbulence, this aspect has either to be omitted in the abstract or also to be addressed in detail.

Specific issues: Abstract: - line 3: "the difficulties encountered" – which difficulties do you mean? It sounds like a reference to some known issue. Please specify if so, otherwise express differently. - line 8: "Uncompensated" – Within the DELICAT setup, a compensating beam steering device was used. - The subject "turbulence" is not addressed again within the whole paper I Introduction: Page1: - lines 17-19: The given references are not related to DELICAT, please refer here to Vrancken et al. Appl. Opt. 2016 - line 19: No, they are not focused - line 21: in future - line 22: air density fluctuation: the right reference would be Feneyrou et al. Appl. opt 2009, or Vrancken et al. Appl. Opt. 2016 - line 23 : "absence of >particle< scatterers" – evidently, without any scatterers (as molecules), there would be no return Page2: - line 1-2: Unclear. What "filtration" are you talking about? Why should it be impossible to cut responses from all "types" of aerosol? Filtration by interferometric methods may reduce the aerosol part to an insignificant level, filtration by molecular (I2, e.g.) or atomic absorption (Cs, e.g.) to an even lower level. This is not addressed in the given references. Veerman et al 2014 (and in more detailed description Vrancken et al. 2016) only use depolarization, which is by no means a filtration method but rather a "hint" to depolarizing aerosols (as the respective authors state). - line 4-5: What do you mean by "signal at the background level"? - line 11 "significant concentration changes in the could split" – phrase fraction – revise! - lines 14-15: "GW also impact CAT" – the relationship between GW and CAT clearly is more complex than just "impacting" – please revise! - line 15: Reference Fritts and Alexander, 2003 refers to the Middle Atmosphere, there are more appropriate references than this one, also by the same author. - line 24: "Strong laser pulses may indicate a non-linear interaction with the medium, resulting in the filamentation effect (Kosareva et al., 2006; Kandidov et al., 2009)." – This effect is of no relevance for lidar, please suppress! - lines 32-33: "Variations of sensing angles for lidars mounted on gyro-platform should be within the error limits of these compensating systems" – unclear what you mean by this. "mounted on" ?? What exactly do you mean by gyro-platforms? Aircraft IRS? This measures the angles, laser-gyros are more precise and accurate than mechanical ones. A compensating platform steers the laser beam in

order to "compensate" the pitch and yaw angles of the aircraft. It also features some "noise", as does the estimation algorithm. This adds up to the fluctuations you refer to. Vrancken et al. 2016 have shown a combination of laser-gyro IRS and mechanical beam steering with an pitch/yaw "noise" of <0.05° rms (within a certain dynamics bandwidth). Page 3: - lines 8-9: I wonder if the Klyatskin references refer to the lower atmosphere, as you state or not to some very general flow. - lines 10-14: This refers to the VERY special case of a volcano eruption – this does not describe the occurrence of such clusters in general. I suggest removing this phrase. Page 4: - line 5: "distinguish impact from the natural changes caused by wind or time evolution. " ?? What do you mean by wind or time evolution" Evolution of what? 2 Observation model and typical scales Page 4: - lines 12-13: The particle >concentration< does not scatter the beam, please revise! Page 5: - line 1: Wrong. The position is defined by x,y ,z. The >attitude< is defined by roll, pitch, yaw. - line 10: References Huffaker and Hardesty, 1996; Inokuchi et al., 2009; have nothing to do with DELICAT. - lines 11-12: "suggest that it may be possible to observe aerosol clusters with evolution time smaller than that of the measurement time interval." Not logic. How could that be possible to observe a phenomenon that is faster than the measurement frequency? Possibly you mean something different, but please revise! - line 14: "imaging" is perhaps not the best notion - line 16: "volume is large Lsin(..)" – omit "large" - line 19: "technical requirement"? for what purpose exactly? Within DELICAT or similar (JAXA Inokuchi, e.g.) the requirement is NOT proper sensing of aerosol clusters. Page 6: - line 10: What do you mean with the max. distance scale Lext? How is it defined by mol. extinction (which depends on mol. scattering)? - line 12: How is Lmax defined? What value is reached by intensity I? - line 15: repeated mention of tau=10ns typical for lidars (cf. line 13) - line 17: the "sensing path" does not end somewhere (Lmax), only the intensity further drops with $L^2$. - line 19: well, usually it is the employed laser technology that sets a limit to the repetition rate. Theoretically, the definition (which is not clear to me) of an Lmax, sets the upper bound for a high rep system. - lines 25ff. Therefore, you state, the used repetition rate should be <10kHz; granted. How does this allow for the assumption of

"delta-pulses" (and what are delta-pulses?), you possibly mean Dirac function, but this assumption has nothing to do with the repetition rate, BUT with the shortness of the pulse. 10ns/1.5m is short in comparison to the considered spatial scales (of turbulence or your clusters) here → thus it is a valid assumption. You conclude another time: "assuming, therefore, the backscattered signals to be independent for each pulse". Again, this has nothing to do with the former, but with lack of coherence between the pulses (if and only if there is no coherent relation between the pulses; true for a Q-switched laser in the case of DELICAT. Not true for some other laser types.). - General: So far, there is no consideration at all, of how fast the sensing laser beam can move from position 1 to 2 within (t2-t1). This is an important topic, since for shorter time scales the residual pitch "noise" of a compensated platform (based on an angle-resolving IRS) surely is lower than when looking on longer time scales. Reasons: The movement of the aircraft itself is not erratically abrupt, but follows its own inertia (so essentially it depends on its mass/distribution). The movements of the compensating platform also have a certain bandwidth. Where is this aspect considered in your study? 3 Modeling of an aerosol cluster lidar image Pages 7 and 8: The derivation of a lidar signal has been published very often, there is no need to perform this in this detail here, I suggest to considerably shorten it. Page 8 - line 14: here you refer to aerosol backscatter but use subscripts M and MB for molecular. Further, in particular for aerosol, there is not a "typical" value. Cf. Vrancken et al. 2016 Table 3 and in particular references therein, or Groß et al. ACP 2013 (DOI: 10.5194/acp-13-2487-2013), e.g. Page 9: - lines 19ff: The discussion of the retrieved lidar signal (Fig. 3) is rather long, to obtain such "bars" (for lifespan> Lmax/u0) is rather evident. Please try to clarify and shorten. 4 The impact of measurement direction fluctuations on cluster lidar images Page 10: - line 2: Again misconception of the term "position". You probably mean "attitude" - line 6: here is the first notion of a time constant of this pitch variation (as mentioned above): Why do you choose 20s? Please explain/justify. - Eq. (7): I do not see any periodicity. Is there lacking a sin/cos? - line 10: Why do you choose 3 and 6°?? On page 2, you talked about a tenth of these values. (apparently, this is a typo) - line 11:

[Figure]

instead of "position", better use "altitude" (if this is what you mean) - Figure4: I do not see the utility of this figure – what does it show me? To what extent does it help the paper to arrive at its conclusion? 5 Airborne lidar measurements in presence of pitch angle fluctuations Page 13: - line 24: A more comprehensive source is Vrancken et al. Appl. Opt. 2016 - line 31: Please use the 24h format, p.m. and a.m. is not used in aviation nor science (as recommended by ISO 8601) Page 14: - line 1: The altitude is 9.46km (flown pressure altitude FL310), NOT the height which is defined differently! - General: You are presenting here lidar data obtained with the DELICAT instrument. o You infer that there is no pitch fluctuation in Fig. 6a. How do you arrive at this conclusion? o What is different to Fig.6b, 10min earlier? Why do you suppose a fluctuation here? o There is no relation to any quantity (pitch fluctuation range, time constant, vertical extent of "cluster" etc.) responsible for these signal variations. Please discuss! - line 11: aerosol cluster backscatter (not reflection!) 6 Conclusions Page 14: - line 16: "are the most important factor for the discussed airborne lidar sensing scenario" – factor in what respect. Phrase is not complete. Page 15: - line 2: "We also show that LIDAR sensing ahead along the flight direction can potentially provide information about aerosol temporal evolution characteristics." – How did you show that? It appears to me, that you primarily showed that the pitch angle fluctuation (if there is such of a sufficient level, with an appropriate time constant/bandwidth) would impede obtaining such information. Between the lines one may also suspect that the "arbitrary" pitch scanning delivers information on the vertical extent of aerosol (cloud?) clusters. Under very rare circumstances (that you should describe), despite pitch fluctuations, one may retrieve information on the temporal evolution. But I understood that this was just the purpose of this paper to derive exactly the necessary conditions. But I do not see them described here. - line 5: The numerical simulations are based on somewhat arbitrary values (pitch values and time constant), so I do not see what to conclude from them. - lines 9-10: "We formulate criteria for distinguishing this impact from the temporal evolution of atmospheric aerosol clouds" – What criteria? I do not find them. If you mean page 13, lines 11-13 – this is rather obvious.

---

## Author Comment (AC3) · 2 Sep 2017

Dear Dr. Vrancken,

Thank you for your comment. The paper was revised in order to clarify our main statements and highlight foundation for our choice of simulating parameters. For this reason, the data of aircraft measurements of pitch angle fluctuations were included in Section 4. The simulation of the interval of the experimental measurements was also included in the Section 5. The discussion in both simulation and experiment results was expanded. The misprints and misleading places were corrected. The line-by-line answer on your comments is presented below.

"*This paper aims at comparing airborne horizontal lidar measurements of aerosol clusters (or possibly hydrometeor clusters / small clouds) is the presence of beam angle fluctuations obtained during an airborne campaign with numerically modelled lidar signals for the same case.*"

That is what we declared and that was done during the research.

"*The declared purpose (in the abstract) is to give a boundary condition in order to separate lidar signal variations attributed to turbulence (molecular density variations) from variations due to beam angle fluctuations*".

I would like to provide a detailed answer on this remark, which reappeared a few times through the comment.

The DELICAT system design, namely measuring of two signals with parallel and perpendicular polarizations correspondingly, allows easily distinguishing the signal from non-spherical aerosol and turbulence. The separation of the signal from spherical aerosol, which does not depolarize the incident radiation, from turbulence has difficulties because both turbulence and symmetrical aerosols have the same polarization properties. The aerosol density fluctuations which are expected in the strong turbulence areas can lead to additional signal variations aggravating the turbulence strength estimate.

However, the question of separation of the turbulence and aerosol signal responses was not the goal of our research. The main problem, upon which we focus in this paper, is strong signal variations, such as presented on Fig.6, which have no explanation. Insight into possible reasons of this behavior allows distinguishing these variations from, for example, the temporal evolution of the aerosol along the aircraft path. In our research, we considered a wide range of aerosol clusters' sizes and cluster evolution times and took into account different positions of these clusters ahead the flight direction. Based on our analysis, we can specify the range of possible sizes of the aerosol clusters that can influence the lidar backscattered signal in the presence of pitch angle fluctuations presented in the figures and discussed in the text. Finally, we conclude that the experimental measurements for specific flights contain signal variations with the same type of behavior, corresponding to areas of strong pitch angle fluctuations measured onboard the aircraft.

The research exposed the signal fragments that was outside of the flight trajectory and cannot be used for the turbulence strength estimation without additional assumptions (added to the Conclusions):

"The signal from the areas with significant pitch angle fluctuations can be used only with additional assumptions due to the fact that the sensing beam deviates from the flight trajectory. We need to assume that turbulence strength does not significantly change at the scale of this deviation which is fulfilled only for the short distances and small angle fluctuations. Otherwise this deviation would lead to the turbulence strength estimation changes which cannot be corrected due to absence of backscattered signal from the

actual aircraft trajectory. At the same time, the aerosol clusters' evolution in absence of significant uncompensated fluctuations of the pitch angle should not prevent to the turbulence strength estimation. The numerical simulations shown that, for reasonable parameter range, these cases can be distinguished."

For the sake of clarity, the fourth sentence in the abstract which could be misleading was changed as follows, in order to better specify the research goals:

"In this paper, we discuss possible error sources imminent to this technique, related to fluctuations of the flight parameters, which may lead to strong signal variations caused by the random deviations of the sensing beam from the forward flight trajectory."

"*The assumptions taken on values of the beam movement are rather arbitrary and not well-founded as are the parameters of the aerosol "clusters""*

The choices of both maximal amplitude of pitch angle fluctuations and cluster's parameters are based correspondingly on the rms of typically used compensation systems and observations of aerosol layers/clusters (detail discussion of quantitative cluster's characteristics with citations is given in Introduction).
In order to make clear our choice of beam movement parameters the new figures (Fig.6 (b) and (d)) presenting on-board measurements of pitch angle fluctuations were incorporated in the paper.

[Figure]

"Fig.6 The experimental data and numerical simulations: (a) the normalized intensity $I_{norm}$ measured during 1 minute at airborne experiments (20.22-20.23); (b) measured pitch angle fluctuations correspond to (a); (c) simulations of the exeriment presented in (a), the clusters are marked by their numbers; (d) the normalized intensity Inorm measured during 1 minute at airborne experiments (20.32-20.33)."

The simulations of the experiment were also performed (Fig.6c). This simulations demonstrate the values of the clusters' characteristics and pitch angle fluctuations which fit the experimental observations. This simulations as well as measured pitch angle fluctuations is now discussed in the experimental section:

"The experimental observations shown in Fig.6 (b) and (d) demonstrate that there are fast and slow pitch angle fluctuations, with the characteristic time scales of 3-4 seconds and about 10-20 sec, respectively. The dotted lines in panels (a), (b) and (b) highlight period of these fluctuations. For the visual convenience only periods of few fast fluctuation in Fig.6 (b) and only few slow fluctuations in Fig.6 (d) are highlighted. The pitch angle fluctuations result in significant changes of backscattered signal. This impact can be seen, for example, in Fig.6a where each signal breach is a result of corresponded pitch angle fluctuations. Two significant signal changes due to slow pitch angle fluctuations can be seen in Fig.6d. The two clusters, first at the $u_0 t_0$=5 km (30 sec) and second at $u_0 t_0$=8 km (50 sec), are suddenly appeared in the field of view due to significant change of the pitch angle presented in the figure by the red curve. In order to resolve the features of backscattered signal caused by the slow pitch angle fluctuations, this type of fluctuations was chosen for the numerical simulation section.

Consider the first and second clusters in Fig.6a which firstly detected at distances 6 km and 14 km, respectively. It can be seen that there are breaches in the signal which appeared simultaneously in both responds. The value of observed signal was decreased by 3 times from the undisturbed value in the breaches. The breaches demonstrate the same behavior as simulated (see Fig.6c) which is typical for the case of presence both uncompensated pitch angle fluctuations and aerosol clusters.

In order to simulate observed in (Fig.6a) effect, we chose four clusters with parameters presented in Table 2. The results of the simulations are presented in Fig.6c. The density of the first and last cluster is twice less than that of the second and third cluster. The first and the second clusters have vertical thickness about 100 meters, while the third and fourth one have thickness about 300 meters. For this reason, the pitch angle effect on the variations of the backscattered signal from the last clusters is weak. The period of pitch angle fluctuations was chosen $T_\varphi$ = 2:85 sec in accordance with observed fast fluctuations (Fig.6a). The correcting parameter $\phi = 0.5\pi$.

 The maximal amplitude $\varphi_0$ of the pitch angle fluctuations in the simulation was 0.6 degree (corresponds to 0.2 degree rms). This parameters of the clusters and pitch angle fluctuations allows to fit the experimentally observed decreasing of the signal level and time interval between the signal reappearance (sizes of the breaches). Based 5 on our numerical analysis we can conclude that the characteristic vertical size of the aerosol clusters provide a noticeable impact on the backscattered signal is about 50-100 meters. The decreasing of vertical size of the aerosol clusters would increase of this estimation.

If we assume that $\varphi_0$ = 0:15 degree (corresponds to 0.05 degree rms), the experimental results can be approximated with the cluster thickness 25-30 m."

"*Regarding the real airborne measurements, the flight data (from the IS, notably) are not taken into account for reference*"

The flight data were analyzed included all the measured parameters, i.e. changing of altitude during the flight, fluctuations of yaw, roll and pitch angles etc. We did not present these data in the paper, because we arrived at the conclusion that it is only pitch angle fluctuation that really matters, given the reasonable sizes of atmospheric aerosol clusters/layers (discussed in the paper). Based on this comment the discussion of changes of the flight parameters information was expanded in the Section 5:

"The experiment shows that the yaw as well as roll angle fluctuations did not exceed pitch angle ones (excluding few times of significant elevations/descent moments during the flight). The altitude changes during the flight, excluding few areas of the significant elevations/descent, did not exceed a value of about ten meters. In accordance with the sensing geometry under discussion and possible sizes of the aerosol clusters, only pitch angle fluctuations can result in noticeable signal changes. The pitch angle fluctuations presented in Fig. 6b corresponded to the lidar backscattered signal presented in Fig.6a; both backscattered signal and pitch angle fluctuations for the other observation interval are shown in Fig.6d. One can see that backscattered signal breaches appeared simultaneously with the pitch angle fluctuations."

*"The individual analysis and the respective comparison are performed rather qualitatively and "by eye", I cite here: page 13, lines 3-4 – "more distorted", "more intense". A quantitative statement is lacking here."*

The goal of our paper was to find a reason of uncontrolled backscattered fluctuations in the presence of aerosol and qualitatively describe it. Since we concluded that this is pitch angle fluctuations in the presence of aerosol clusters, we distinguished ranges of both possible angle fluctuations and cluster characteristics (sizes, evolution times) which can result in noticeable signal level changes. Thus the general conclusion from Fig.5 is that aerosol cluster's thickness in the range [100-300] m will make a significant impact on the signal level variations for the considered range of correction errors [0.1-0.2] degree rms. This conclusion is based on the quantitative analysis and contains value of parameter ranges.

Following your comment, the additional discussion of intensity variations was added:

"For this reason the images are more distorted at the right side of each pane of Fig. 5.

For example, the backscattered signal at the sensing distance $L$=6 km at the time corresponding to aircraft trajectory coordinate $u_0t_0$=5 km in presence of aerosol clusters with thickness $2\delta o$=100 m (Fig.5(a,b)) decreased by about 20% (60%) from the level without the pitch angle fluctuations (see Fig.3) for $\varphi_0$=0.3 degree and $\varphi_0$=0.6 degree respectively. The signal decreased by about 10% in the presence of aerosol clusters with thickness 300 m for the $\varphi_0$=0.6 degree and it had no noticeable changes for larger vertical sizes of cluster or smaller angles (Fig.5(c,d)). The backscattered signal from the aerosol layer at the sensing distance $L$=15 km at the time corresponded to aircraft trajectory coordinate $u_0t_0$=5 km with thickness 100 m decreased by about 85% for the $\varphi_0$=0.3 degree and absent (only background level) for the $\varphi_0$=0.6 degree. The signal decrease about 35% and 45% in presence of aerosol clusters with thickness 300 m for the $\varphi_0$=0.3 degree and $\varphi_0$=0.6 degree correspondingly, while for the thickness of the cluster about 900 m the only noticeable change (about 12%) can be found for the $\varphi_0$=0.6 degree. Similar effects can be found in Fig.5 for each other moment of time (corresponding to flight trajectory coordinate $u_0t_0$)."

"The vertical beam deviation caused by pitch angle fluctuations is about 30 m and 60 m at 6 km distance for maximal amplitude of angle fluctuations $\varphi_0$=0.3 degree and 0.6 degree, respectively (Fig.4). It increases up to 75 and 150 meters for the 15 km distance. The sensing beam can easily move outside the aerosol cluster with a thickness less than the doubled shift size. Even for a movement with a smaller amplitude the backscattered signal will decrease due to decreasing of the cluster density nearby its edge."

*"The respective chapters 4 and 5 are very short in content and discussion"*

Following your previous comments discussion in both these sections was expanded.

"*The conclusion effectively is none, because there is no quantitative measure that has been determined*"

The estimation of range of aerosol cluster thicknesses for possible IRS system rms is given in the Conclusions. Since it was the main problem stated in abstract and discussed through the paper, this estimation is the main quantitative result of the paper. Following this comment, the additional estimations are included in the Conclusions:

"The lidar backscattered signal from 15 km sensing distance can disappear (or decrease by about 85%) for compensation of pitch angle fluctuations with 0.2 degree rms (0.1 degree rms) in presence of aerosol clusters with characteristic vertical scale about 100 meters. Aerosol clusters with characteristic vertical scale about 300 meters lead to 45% (35%) signal decreasing for the same sensing distance and pitch angle fluctuations. The signal level fluctuations about 60% (20%) can be caused by pitch angle fluctuations with 0.2 degree rms (0.1 degree rms) at the 5 km sensing distance. Pitch angle fluctuations in presence of aerosol clusters with [100-300] m with angular correction smaller than 0.2 degree rms lead to breaches in the backscattered signal.

Presence of two or more aerosol clusters allows to easily distinguish the areas of significant beam wander due to signal decreasing caused by pitch angle fluctuations."

In accordance your previous comments the numerical simulation of the experiment was performed. This result is also mention in the Conclusion:

"Simulations of the experiment are performed with assuming the aerosol clusters thickness about 100 meters (1000 m for the large cluster) for the case of pitch angle compensation with 0.2 degree rms. For compensation with 0.05 degree rms noise the corresponded value of the aerosol clusters' thickness 25-30 meters (about 250 meters for the large cluster)."

"*The only conclusion that is valid is that such pitch variations of the sensing beam does generate fluctuations in the (normalized) signal when encountering a vertically extended object. But this is geometrically obvious.*"

Our conclusion about characteristic range of both pitch angle fluctuations and sizes of aerosol clusters which can significantly impact on the lidar signal have a solid foundation in presented simulations and analysis presented in the text. According to your previous comment the experimental measurements of pitch angle fluctuations were included in the paper for clarification of our choice of simulation parameters. The discussion of both experimental and simulation results was expanded.

"*The promised conclusion on separation of turbulence signal fluctuation (which is on the sub-percent level, Vrancken et al. 2016) from pitch/cluster variation is not given since mere arbitrary color bars are given without any notion on temporal characteristics.*"

This was not a goal of our research. The sentence in the abstract which could mislead to this conclusion was changed (the detail answer is presented on the first page).

"*In addition to above described contentual shortcomings, the text also yields many conceptual mishaps, some of them described here below. Some of them may be founded in language problems, other in hastily written assertions*"

The text was revised based on your comments.

"*In summary, I suggest at least a major revision of this document with not only incorporating the points listed here, but also delivering resilient numbers. This should include a thourough study and analysis of a whole range of possible pitch variations (amplitude/time constant-bandwidth), cluster vertical extent, cluster scattering/concentration distribution and temporal evolution of the latter. This was promised in the abstract and introduction.*"

The study and analyses of the pitch angle fluctuation effect on lidar backscattered signal was announced and performed. Based on your other comments the additional discussion of values presented in Figures were included in the paper. Our model was also used for simulation of the experimental observations (Fig.6c). Below, I shortly review the main paper results.

In this research, we, based on the cited literature, considered a wide range of clusters' sizes and possible evolution times. In the paper we only presented a range of cluster' parameters which can impact on the lidar signal based typical rms values of the compensation systems for the angle fluctuations. The period of the pitch angle fluctuations was based on experimental observations (Fig.6(b,c)).

The clusters' vertical sizes were limited by the range from 100 to 900 meters because the discussed effect disappears for larger sizes in the considered range of possible pitch angle fluctuations. This range was based on the typical errors of compensation systems used in the civil aviation. Maximal amplitude of pitch angle fluctuations changed from zero up to 0.3 degrees (0.6 degrees) which correspond correction with 0.1(0.2) degree rms. The on-board measurements of fluctuations of the pitch angle demonstrate the values about 1-2 degrees. We have no information about the success of the compensation, but the effects (contaminated with the pitch angle fluctuations) are clearly seen in the Figure 6. Therefore, the pitch angle fluctuations were only partially compensated.

We considered the range of aerosol clusters' horizontal sizes from 0.5 up to 2 kilometers. A larger size of clusters results in a wider signal area and does not produce any specific effects. Smaller cluster sizes are highly unlikely. We also considered the time of cluster temporal evolution to be in the range from 10 sec up to 60 sec. It is obvious that the evolution time can be much larger than 1 minute. However, the clusters with such a slow evolution look like time-constant structures during the experimental 1-minute observation. The faster evolution of clusters is doubtful.

Thus, we considered all reasonable ranges for pitch angle fluctuations, time of the clusters evolutions and their horizontal and vertical sizes. Following one of the previous comments, the simulations of the experiment was performed. The simulations showed that the ranges of aerosol clusters' parameters and pitch angle fluctuations, considered in the paper, allow to successful simulation of the effect which was observed experimentally.

"*Regarding turbulence, this aspect has either to be omitted in the abstract or also to be addressed in detail.*"

The misleading sentence in the abstract was changed.

You also mentioned the problem: (page 6, line 25): "*Because the shot frequency is taken to be less than 1=tmax, we may use the approximation of delta-pulses, assuming, therefore, the backscattered signals to be independent for each pulse.*"
These sentences were changed following your comments below in the "Specific issues" for the page 6.

Answer for "Specific issues" section

 "*Abstract:*
*- line 3: "the difficulties encountered" – which difficulties do you mean? It sounds like a reference to some known issue. Please specify if so, otherwise express differently.*"

We referred to signal variations such as presented in the Fig. 6. This sentences was changed:

"However, the strong variations of signal level sometimes, which were observed during the DELICAT measurements but not explained, indicated the need of a better understanding the observational errors due to geometrical factors"

""*Uncompensated" – Within the DELICAT setup, a compensating beam steering device was used.*"

We meant the part of the signal which was not compensated (due to error limits of the using system). The sentence was changed:

"The part of pitch angle fluctuations uncompensated by the beam steering device in the presence of aerosol concentration variations can lead to noticeable signal variations that can be mistakenly attributed to wind shear, turbulence or fast evolution of aerosol layer."

"*The subject "turbulence" is not addressed again within the whole paper*"

We focus on the unexplained strong signal variations observed in the airborne sensing. The impact of pitch angle fluctuations on the turbulence estimation was shortly formulated according to your comment (the detail answer is presented on the first page).

"*Introduction: Page1:*
*- lines 17-19: The given references are not related to DELICAT, please refer here to Vrancken et al. Appl. Opt. 2016 - line 19*"

These sentences were reformulated. Instead "Recently, a medium range lidar was developed, built and tested in the framework of the DELICAT (DEmonstration of LIdar based Clear Air Turbulence detection)" it says now:

"Recently, a medium range lidars were developed, built and tested [References]. One of these systems was developed in the framework of the DELICAT project (DEmonstration of LIdar based Clear Air Turbulence detection) [*Vrancken et al. Appl. Opt. 2016*]"."

"*line 19: No, they are not focused*"

The word "focused" was changed on "are designed to work up to":

"Medium range systems are designed to work up to 20-30 km sensing distance, which corresponds to 2–10 minutes of warning time for typical flight speed of airplane and helicopter correspondingly."

"*line 21: in future*"

This misprint was eliminated.

"*line 22: air density fluctuation: the right reference would be Feneyrou et al. Appl. opt 2009, or Vrancken et al. Appl. Opt. 2016*"

The cited work is also discuss possibility of turbulence sensing based on air density fluctuation. Following your comment the references were added.

*"line 23: "absence of >particle< scatterers" – evidently, without any scatterers (as molecules), there would be no return"*

This misleading term was changed on the "aerosol scatterers".

*Page2:*
*- line 1-2: Unclear. What "filtration" are you talking about? Why should it be impossible to cut responses from all "types" of aerosol? Filtration by interferometric methods may reduce the aerosol part to an insignificant level, filtration by molecular (I2, e.g.) or atomic absorption (Cs, e.g.) to an even lower level. This is not addressed in the given references. Veerman et al 2014 (and in more detailed description Vrancken et al. 2016) only use depolarization, which is by no means a filtration method but rather a "hint" to depolarizing aerosols (as the respective authors state)."*

The sentences were changed:

"The signal filtration is a good method to exclude undesirable contributions. For example, Hair and co-authors used an extremely narrowband iodine vapor (I2) absorption filter to eliminate the aerosol returns and pass the wings of the molecular spectrum (Hair et al., 2008). At the same time, in the DELICAT system the depolarization was used (Vrancken et al. 2016)."

*"- line 4-5: What do you mean by "signal at the background level"? "*

This sentence was changed and the misleading term is excluded:

"The presence of atmospheric aerosol should not be a critical problem for turbulence detection."

*"line 11 "significant concentration changes in the could split" – phrase fraction – revise!"*

The sentenced was revised:

"Clouds could be split up into numerous small clusters at the horizontal scale of one or several kilometers. Such splitting was observed for different types of aerosol."

*"lines 14-15: "GW also impact CAT" – the relationship between GW and CAT clearly is more complex than just "impacting" – please revise!"*

The sentenced was revised:

"Gravity waves are one of the reasons of CAT and new results suggest that turbulence was most strongly forced at the scale of about 700 m."

*"line 15: Reference Fritts and Alexander, 2003 refers to the Middle Atmosphere, there are more appropriate references than this one, also by the same author."*

The paper (Fritts and Alexander, 2003) contains information for the altitude range from 10 to 110 km. Since the flight altitude is about 10 km at the longest part of the typical aircraft flight, we believe that (Fritts and Alexander, 2003) is appropriate here. The second work which we cited in this place is (Nappo, 2013) which is a book and contains a lot of information about gravity waves in wide altitude range.

*"line 24: "Strong laser pulses may indicate a non-linear interaction with the medium, resulting in the filamentation effect (Kosareva et al., 2006; Kandidov et al., 2009)." – This effect is of no relevance for lidar, please suppress!"*

This discussion was excluded.

*"lines 32-33: "Variations of sensing angles for lidars mounted on gyro-platform should be within the error limits of these compensating systems" – unclear what you mean by this.*

*"mounted on" ?? What exactly do you mean by gyro-platforms? Aircraft IRS? This measures the angles, laser-gyros are more precise and accurate than mechanical ones. A compensating platform steers the laser beam in order to "compensate" the pitch and yaw angles of the aircraft. It also features some*

*"noise", as does the estimation algorithm. This adds up to the fluctuations you refer to. Vrancken et al. 2016 have shown a combination of laser-gyro IRS and mechanical beam steering with an pitch/yaw "noise" of <0.05° rms (within a certain dynamics bandwidth)."*

We take into account the typical range of the laser-gyros (0.1-0.2 rms) which is close to DELICAT gyros compensation error (0.05 degrees rms). We discuss values of the angle fluctuations which are in the limits of compensation errors. Following your comment the sentence was included in the numerical simulations' discussion (section 4):

"Lidar images computed in the presence of pitch angle fluctuations in the typical range of the laser-gyros (0.1-0.2 rms) are presented in Fig. 5."

Following your comment the angle compensation with the 0.15 degree rms was considered in the simulations. The following sentence was added to the Section 5:

"If we assume that '0 = 0.15 degree (corresponds to 0.05 degree rms), the experimental results can be approximated with the cluster thickness 25-30 m."

Since such aerosol cluster's thickness is not commonly observed, the values of the standard rms of the giro-systems (0.1-0.2 degree rms) were hold in the numerical simulation section with the corresponding vertical sizes of the aerosol clusters (100-300 m).

*"Page 3:*
*- lines 8-9: I wonder if the Klyatskin references refer to the lower atmosphere, as you state or not to some very general flow."*

The Klyatskin theory can be applied to the lower atmosphere (source: cited books and personal conversation of the authors with Dr. Klyatskin in 2015).

*"- lines 10-14: This refers to the VERY special case of a volcano eruption – this does not describe the occurrence of such clusters in general. I suggest removing this phrase."*

The aerosol clusterization depends on atmospheric conditions (wind, humidity, pressure etc.) and aerosol layer parameters. Even though the discussed aerosol is specific (volcanic), it still presents an example of an aerosol cluster and typical at least for the volcanic aerosol. At the same time we considered a lot of different aerosol types in Introduction which presents other examples of aerosol clusters with similar parameters.

*"Page 4:*
*- line 5: "distinguish impact from the natural changes caused by wind or time evolution. " ?? What do you mean by wind or time evolution" Evolution of what?"*

This place was changed: "distinguish of pitch angle fluctuation impact from the evolution of aerosol clusters".

*"2 Observation model and typical scales*

*Page 4:*
*- lines 12-13: The particle >concentration< does not scatter the beam, please revise!"*

The sentence was changed:

"The laser beam scatters on thermodynamic fluctuations of air density (Fabelinskii, 2012) and particles of solid or liquid aerosol (Bohren and Huffman, 2008)."

*"Page 5:*
*- line 1: Wrong. The position is defined by x,y ,z. The >attitude< is defined by roll, pitch, yaw."*

The sentence changed:

"The fluctuations of the sensing direction during the flight can be defined by fluctuations of three angles: roll, yaw and pitch."

"*line 10: References Huffaker and Hardesty, 1996; Inokuchi et al., 2009; have nothing to do with DELICAT.*"

This misprint was corrected.

"*lines 11-12: "suggest that it may be possible to observe aerosol clusters with evolution time smaller than that of the measurement time interval." Not logic. How could that be possible to observe a phenomenon that is faster than the measurement frequency? Possibly you mean something different, but please revise!*"

When we said "measurement time interval" we meant the time interval during which the cluster can be observed until the aircraft flight over it. This time is defined as maximal sensing distance divided by the aircraft speed. The evolution time is defined as a time during which a cluster significantly changes (Eq. 6 provides exact definition). Thus evolution can be seen if the observation time is larger than interval of time changes. Otherwise the layer/cluster looked as having constant characteristics. In order to avoid misunderstanding the sentence was revised:

"Airborne lidar measurements in the flight direction suggest that it may be possible to observe evolution of the aerosol clusters with evolution time smaller than the observation time."

"*- line 14: "imaging" is perhaps not the best notion*"

The "imaging" was changed on "lidar observations":

"In this paper, we simulate and discuss the influence of airplane pitch angle variations on the lidar backscattered signal from the aerosol clusters."

"*- line 16: "volume is large Lsin(..)" – omit "large"*"

The "large" was omitted.

"*- line 19: "technical requirement"? for what purpose exactly? Within DELICAT or similar (JAXA Inokuchi, e.g.) the requirement is NOT proper sensing of aerosol clusters.*"

The sentence was reformulated:

"In order to avoid the signal variations caused by the pitch angle fluctuations the condition $L\sin(\varphi/\delta_0)<1$ on the maximal acceptable beam angle deviation $\varphi$ should be fulfilled in presence of aerosol clusters with vertical size about $\delta o$."

"*Page 6:
- line 10: What do you mean with the max. distance scale $L_{ext}$? How is it defined by mol. extinction (which depends on mol. scattering)?*"

We meant that the molecular extinction distance $L_{ext}$ is longer than the maximal sensing distance $L_{max}$. The $L_{ext}$ was moved from the end of the sentence:
"Assuming that the molecular scattering is negligibly weak and neglecting molecular absorption, we may accept the length of molecular extinction $L_{ext}$ to be the maximum distance."

"*- line 12: How is $L_{max}$ defined? What value is reached by intensity I?*"

According to your comment the following explanation is included in the text:
"Distance $L_{max}$ is defined as the maximal distance for which we are still able to register backscattered signal. Specifically, in our simulations we limited by signal registered with time delay corresponded to 16 km distance suppose that the signal from longer distance cannot be registered due to the noise."

"*line 15: repeated mention of tau=10ns typical for lidars (cf. line 13)*"

The repeated information was excluded.

*"line 17: the "sensing path" does not end somewhere (Lmax), only the intensity further drops with L².""*

We define the sensing path as the path during which the experimental equipment is able to register backscattered lidar signal. Thus, based on our definition, the sensing path ends when we cannot resolve backscattering signal from the background light or errors of signal registration. The sentence was change in order to avoid misunderstanding:

"We define the sensing path as the path during which the experimental equipment registers the backscattered lidar signal."

*"- line 19: well, usually it is the employed laser technology that sets a limit to the repetition rate. Theoretically, the definition (which is not clear to me) of an $L_{max}$, sets the upper bound for a high rep system."*

Following your comment the sentences was rewritten:

"For the value of $t_{max}$ is about 0.1 milliseconds, the distance $L_{max} = 15$ km."

*"lines 25ff. Therefore, you state, the used repetition rate should be <10kHz; granted.*

*How does this allow for the assumption of "delta-pulses" (and what are delta-pulses?), you possibly mean Dirac function, but this assumption has nothing to do with the repetition rate, BUT with the shortness of the pulse. 10ns/1.5m is short in comparison to the considered spatial scales (of turbulence or your clusters) here –> thus it is a valid assumption. You conclude another time: "assuming, therefore, the backscattered signals to be independent for each pulse". Again, this has nothing to do with the former, but with lack of coherence between the pulses (if and only if there is no coherent relation between the pulses; true for a Q-switched laser in the case of DELICAT. Not true for some other laser types.)."*

This part is revised according your comment:

"Based on the absence of coherent relation between pulses we assume the backscattered signals to be independent for each pulse."

"Because the lidar pulse is short (10 ns) in comparison to the considered spatial scales, we can use the Dirac function which significantly simplifies the analytical solution."

*"- General: So far, there is no consideration at all, of how fast the sensing laser beam can move from position 1 to 2 within ($t_2$-$t_1$). This is an important topic, since for shorter time scales the residual pitch "noise" of a compensated platform (based on an angle-resolving IRS) surely is lower than when looking on longer time scales. Reasons: The movement of the aircraft itself is not erratically abrupt, but follows its own inertia (so essentially it depends on its mass/distribution). The movements of the compensating platform also have a certain bandwidth. Where is this aspect considered in your study?"*

This aspect was considered in our study in the framework of presented equations and shown in Fig.4. Fig. 4 presents the dependence of laser beam movement from the aircraft trajectory as a function of time and distance from the aircraft. The time was shown as $u_0t$ (distance of aircraft flight). Taking into account the discussed aircraft velocity (170 m/sec) and distance (20 km) the whole observation time here is evaluated to about 2 minutes. According to your comment, the discussion of the results presented in Fig. 4 is expanded:

"The distance $u_0t_0$=20 km corresponds to 2 minutes of airborne observation (for aircraft speed 170m/sec). Beam displacement is changing from 0 to 160 m (from 0 to 80 m) for $\varphi_0$=0.3 degree ($\varphi_0$=0.6 degree) for about of 10 sec. The speed of movement was defined by period $T_\varphi$=20 sec".

This discussion also answered one of your comments below about the utility of Figure 4.
According your previous comment the another $T_\varphi$=2.8 sec was considered (Fig.6c).

*"3 Modeling of an aerosol cluster lidar image*
*Pages 7 and 8: The derivation of a lidar signal has been published very often, there is no need to perform this in this detail here, I suggest to considerably shorten it."*

We did not present the detailed derivation in the paper. We only shown Eq. (3) for the intensity of the lidar signal (with citation) and give the explanation of the terms (Eq.4 and text around). This takes about half of the page and helps the readers who are not well familiar with the topic. The final Eq. 5 is needed for understanding of images represented on Figures 3, 5, 6 and cannot be eliminated.

*"Page 8*
*- line 14: here you refer to aerosol backscatter but use subscripts M and MB for molecular.*

*Further, in particular for aerosol, there is not a "typical" value. Cf. Vrancken et al. 2016 Table 3 and in particular references therein, or Groß et al. ACP 2013 (DOI: 10.5194/acp-13-2487-2013), e.g."*

Indeed the aerosol values discussed in (Ishimary, 1978) correspond to weak water aerosol and can be considered as typical only for this case. The sentence was changed and citation on your paper was included:

"This value typically corresponds to weak water aerosol in accordance with Fabelinskii (2012); Ishimaru (1978), which implies that the aerosol scattering does not significantly decrease the propagating laser pulse energy. The values for the other types of aerosol can be found, for example, in (Vrancken et al. 2016)."

*"Page 9:*
*- lines 19ff: The discussion of the retrieved lidar signal (Fig. 3) is rather long, to obtain such "bars" (for lifespan> Lmax/u0) is rather evident. Please try to clarify and shorten."*

The importance of these bars is that they provide non-disturbed signals from the clusters with different time of living and sizes. This bar is needed for understanding the results presented in Fig. 5 though comparing distorted and undistorted signals. In order to make the discussion shorter the sentences contains the repeating information were merged. The following sentences were excluded:

"Vertical cluster scale does not influence the lidar image under the assumed conditions."

"This parameter is important for obtaining images of short-living clusters at large distances L, because the observer moves closer to the detected cluster."

*"4 The impact of measurement direction fluctuations on cluster lidar images*

*Page 10:*

*- line 2: Again misconception of the term "position". You probably mean "attitude""*

The term was corrected.

*"- line 6: here is the first notion of a time constant of this pitch variation (as mentioned above):*

*Why do you choose 20s? Please explain/justify."*

According your comment the following discussion is included in the paper:

"Our choice of the $T_\varphi$ is based on one of the characteristic times of pitch angle fluctuations measured in the experiment. These times vary in the range from few to tens of seconds (Fig.6 (b) and (d)). The considered effect do not disappear for smaller/larger times; such changes would result in changing of thickness of the breaches."

*"- Eq. (7): I do not see any periodicity. Is there lacking a sin/cos?"*

The misprint was corrected.

*"- line 10: Why do you choose 3 and 6°?? On page 2, you talked about a tenth of these values. (apparently, this is a typo)"*

The misprint was corrected.

*"- line 11: instead of "position", better use "altitude" (if this is what you mean)"*

The term was changed.

*"- Figure4: I do not see the utility of this figure – what does it show me? To what extent does it help the paper to arrive at its conclusion?"*

This figure visually represents the value of beam shift along the sensing distance. It helps understanding the difference of the pitch angle effect for different distances. This Figure helps understanding the behavior of lidar signal (which is shown in Fig.5).

*"5 Airborne lidar measurements in presence of pitch angle fluctuations
Page 13:
- line 24: A more comprehensive source is Vrancken et al. Appl. Opt. 2016"*

The reference to this paper was added.

*"- line 31: Please use the 24h format, p.m. and a.m. is not used in aviation nor science (as recommended by ISO 8601)"*

The time format was changed.

*"Page 14:
- line 1: The altitude is 9.46km (flown pressure altitude FL310), NOT the height which is defined differently!"*

The term was corrected.

*"- General: You are presenting here lidar data obtained with the DELICAT instrument.
You infer that there is no pitch fluctuation in Fig. 6a. How do you arrive at this conclusion?"*

We meant that pitch angle fluctuations are in the range of [-1.0,1.0] degrees for the lidar backscattered signal presented in the Fig.6a, while they are about [-0.5,0.5] for the case presented in the Fig.6d, i.e. twice smaller. But they still can affect the lidar backscattered signal. Since the discussion of the experimental results was expanded following your other comments this information is now presented in the paper.

*"What is different to Fig.6b, 10min earlier? Why do you suppose a fluctuation here?"*

We have to note that, based on the flight map, area of the flight was almost behind the Alpes. For this flight, the changes of flight parameters can happen very fast.

*"There is no relation to any quantity (pitch fluctuation range, time constant, vertical extent of "cluster" etc.) responsible for these signal variations. Please discuss!"*

The discussion is added based on your previous comments.

*"- line 11: aerosol cluster backscatter (not reflection!)"*

The term was corrected.

*"Conclusions
Page 14:
- line 16: "are the most important factor for the discussed airborne lidar sensing scenario" – factor in what respect. Phrase is not complete."*

The phrase is rewritten:

"It is shown that the pitch angle fluctuations are the important parameter for the airborne lidar sensing ahead the flight direction in the case when their uncompensated value result in the sensing beam shift about the vertical size of the aerosol clusters."

*"Page 15:*
*- line 2: "We also show that LIDAR sensing ahead along the flight direction can potentially provide information about aerosol temporal evolution characteristics." – How did you show that? It appears to me, that you primarily showed that the pitch angle fluctuation (if there is such of a sufficient level, with an appropriate time constant/bandwidth) would impede obtaining such information. Between the lines one may also suspect that the "arbitrary" pitch scanning delivers information on the vertical extent of aerosol (cloud?) clusters. Under very rare circumstances (that you should describe), despite pitch fluctuations, one may retrieve information on the temporal evolution. But I understood that this was just the purpose of this paper to derive exactly the necessary conditions. But I do not see them described here.*"

This conclusion is based on the results of our numerical analysis. We show that the   In order to make this statement clear, the following discussion of the simulations' results was added to the text around Fig.5:

"As shown in Fig.5, even the clusters with smallest evolution time corresponding to a living time below 30 sec still appeared twice for strongest fluctuations (0.6 degree) for the largest sensing distance. It means that we can observe evolution of the smallest considered cluster (0.5 km length) with the smallest considered evolution time at the any considered sensing distance. The evolution of the cluster is clearly seen in decreasing the signal in the periods between the breaches caused by pith angle fluctuations. Such decreasing can be seen for all considered clusters with and without pitch angle effects (Fig.5)."

This were summarized in the conclusion:

"We also show in numerical simulations that lidar sensing ahead the flight direction can potentially provide information about aerosol temporal evolution characteristics even in presence of pitch angle fluctuations for reasonable cluster size and evolution time at the considered sensing distance."

*"- line 5: The numerical simulations are based on somewhat arbitrary values (pitch values and time constant), so I do not see what to conclude from them.*"

The answer to this comment was given above. Shortly, in order to clarify the choice of pitch angle fluctuations parameters the experimental measurements was included in the Fig.6. The other parameters are based on references which can be found in Introduction with the quantitative characteristics of experimentally observed cluster parameters. One of the conclusion statements is:

"Numerical simulations predict that uncontrolled fluctuations can result in signal noise including extreme fades and spikes. We show that the aerosol concentration variations on a scale of 100-300m have a significant impact on the backscattered signal, if the correction for the angular fluctuation has a residual rms error about of 0.1–0.2 degrees, which is typical for conventional gyro-platforms used in the civil aviation."

*"- lines 9-10: "We formulate criteria for distinguishing this impact from the temporal evolution of atmospheric aerosol clouds" – What criteria? I do not find them. If you mean page 13, lines 11-13 – this is rather obvious.*"

We meant the lines 8-10:
"It may be expected that a natural process intensity, like aerosol evolution due to evaporation or condensation, varies for different clusters. A distortion due to flight direction fluctuations has the same impact on the images of all the clusters observed at the same distance."

This idea is very simple, but taking into account formulated limitations for the cluster parameters and pitch angle fluctuations, it allows to easily distinguish the pitch angle fluctuations. It is not the main result of our paper, however it is important for the analysis of the backscattered signal and distinguishing areas of wandered beam which cannot be used for turbulence strength estimations.